# Human fetal kidney organoids model early human nephrogenesis and Notch-driven cell fate

Michael Namestnikov[1,2,3], Osnat Cohen-Zontag[1,3], Dorit Omer[1], Yehudit Gnatek[1], Sanja Goldberg [iD][1], Thomas Vincent[4], Swati Singh[5], Yair Shiber[6], Tal Rafaeli Yehudai [iD][6], Hadas Volkov[7], Dani Folkman Genet [iD][8], Achia Urbach[8], Sylvie Polak-Charcon[9], Igor Grinberg[9], Naomi Pode-Shakked [iD][2,10], Boaz Weisz[2,11], Zvi Vaknin[6], Benjamin S Freedman[4] & Benjamin Dekel [iD][1,2,3][✉]

## Abstract

Pluripotent stem cell (PSC)–derived kidney organoids are used to model human renal development and disease; however, accessible models of human fetal development to benchmark PSC-derived organoids remain underdeveloped. Here, we establish a chemically defined, serum-free protocol for prolonged culture of human fetal kidney-derived organoids (hFKOs) in vitro. hFKOs self-organize into polarized renal epithelium, reinitiate from NCAM1[+] progenitors, and recapitulate nephrogenic and ureteric bud lineages. Bulk transcriptomics, single-cell RNA sequencing, pseudotime analysis, and immunostaining revealed diverse renal tissue cell populations, with a preserved epithelial progenitor pool and tubular differentiation axis. hFKOs were enriched for Notch signaling genes, enabling single-cell analysis of pharmacological Notch inhibition. This revealed a maturation block with increased nephron progenitors and a shift toward distal over early proximal tubule fates. We also identified a novel prominin-1-expressing cell state that evades Notch inhibition to generate both proximal and distal tubules. Overall, hFKOs provide a faithful model to gain insights into human kidney development, advancing the fields of stem cell biology and regenerative medicine.

**Keywords** Human Fetal Kidney; Kidney Organoids; Nephrogenesis; Notch Pathway; Single-cell Transcriptomics
**Subject Categories** Development; Methods & Resources; Stem Cells & Regenerative Medicine

## Introduction

The mammalian kidney originates from reciprocal interactions between the metanephric mesenchyme and the ureteric bud during fetal development. The metanephric mesenchyme, which contains the nephron progenitor cells (NPCs), is invaded by the ureteric bud, causing the mesenchyme to condense into the cap mesenchyme around the tips of the branching ureteric bud (Kobayashi et al, 2008). Subsequently, the cap mesenchyme differentiates into pre-tubular aggregates, which further differentiate into renal vesicles (RVs) and comma- and then S-shaped bodies (CSBs and SSBs), culminating in fully developed nephrons encompassing glomerular and tubular compartments that drain into the ureter (Little et al, 2007). Understanding the factors governing kidney development is key to uncovering the pathological mechanisms underlying low nephron endowment (Cebrian et al, 2014) and congenital abnormalities of the kidney and urinary tract (CAKUT), which later in life increase the risk of chronic kidney disease and kidney failure (Schedl, 2007; Romagnani et al, 2017). Traditionally, genetically modified mouse models, which are robust yet laborious and low in throughput, have been used to map factors contributing to the patterning and maturation of nascent nephrons and study the deleterious effects of mutations affecting key signaling pathways such as Notch (Kamath et al, 2013) and WNT (Wang et al, 2018b).

Utilization of design principles derived from mouse models and manipulation of human fetal kidney (hFK) tissues has been instrumental in promoting kidney research. This includes the understanding of post-transplant organogenesis, immunogenicity, and vascularization of hFK tissues and grafts of varying gestational age—both whole rudiments and tissue fragments (Dekel et al, 2002b, 2002a, 1997). Other studies have pursued prospective human fetal tissue-progenitor isolation and characterization to identify means of kidney cell therapy (Harari-Steinberg et al, 2013) and short-term cultures to study progenitor cell relationships and

[1]The Pediatric Stem Cell Research Institute and Pediatric Nephrology Division, Edmond and Lily Safra Children's Hospital, Sheba Medical Center, Tel HaShomer, Israel. [2]Gray Faculty of Medical & Health Sciences, Tel Aviv University, Ramat Aviv, Tel Aviv 69978, Israel. [3]Sagol Center for Regenerative Medicine (SCRM), Tel Aviv University, Ramat Aviv, Tel Aviv, Israel. [4]Department of Medicine, Division of Nephrology, Institute for Stem Cells and Regenerative Medicine, and Kidney Research Institute, University of Washington, Seattle, WA, USA. [5]Department of Molecular Genetics, Weizmann Institute of Science, Rehovot 7610001, Israel. [6]Department of Obstetrics and Gynecology, Assaf Harofeh Medical Center, Zerifin, Israel. [7]The Edmond J. Safra Center for Bioinformatics, Tel-Aviv University, Tel Aviv, Israel. [8]The Mina and Everard Goodman Faculty of Life Sciences, Bar-Ilan University, Ramat Gan, Israel. [9]Pathology Department, Sheba Medical Center, Tel HaShomer, Israel. [10]Pediatric Nephrology Unit, Dana Dwek Children's Hospital, Tel Aviv Medical Center, Tel Aviv, Israel. [11]Institute of Obstetrical and Gynecological Imaging, Department of Obstetrics and Gynecology, Sheba Medical Center, Tel HaShomer, Israel. [✉]E-mail: binyamin.dekel@sheba.health.gov.il; Benjamin.dekel@gmail.com

human kidney developmental disease (Pode-Shakked et al, 2016, 2017; Vivante et al, 2013). Importantly, additional groups have described methods for sorting NPCs from human fetal kidney tissue and propagating them in 2D (Brown et al, 2015) and 3D (Li et al, 2016) for ex vivo manipulation and study.

The recent exciting ability to generate hFK-like tissue via differentiation of pluripotent stem cells and principles of organoid science has challenged the need for experimental hFK research (Little and Combes, 2019). To this end, conditions needed for nephron (Taguchi et al, 2014; Morizane et al, 2015; Takasato et al, 2015; Freedman et al, 2015) and collecting system (Zeng et al, 2021) development have been recapitulated to differentiate pluripotent stem cells into kidney organoids (PSC-KOs), providing a valuable tool for disease modeling and drug safety testing and potentially a cell source for kidney regenerative medicine in the form of cell therapy or whole-organoid grafting (Gupta et al, 2020; Freedman, 2015; Freedman and Dekel, 2023). However, single-cell RNA sequencing (scRNA-seq) analysis of these PSC-derived organoids reveals major hurdles remaining, including incomplete maturation (Wu et al, 2018), reproducibility between induced pluripotent stem cell (iPSC) lines (Subramanian et al, 2019), and a variety of unwanted cell lineages (Nam et al, 2019).

Therefore, despite the inherent difficulties of obtaining fresh hFKs and the ethical dilemmas associated with their collection, hFKs remain the gold standard for studying human kidney development in vitro. Indeed, significant efforts have been made to generate reference transcriptional atlases of mid-gestation human fetal kidneys (Lindström et al, 2021; Hochane et al, 2019; Wu et al, 2022). Importantly, leveraging the ability of the adult mammalian kidney to clonally expand mature cell types (Rinkevich et al, 2014), tissue-derived adult kidney organoids/tubuloids have been successfully generated, recapitulating adult tissue growth and repair, allowing the derivation of a personalized kidney platform for disease modeling (Schutgens et al, 2019). We speculated that combining robust organotypic culture technology with human fetal kidney as a cell source would allow us to create an ex vivo 3D biological entity that encompasses the developmental hierarchy present in the native tissue for multiple passages. We utilized a previously described modified medium (hNPSR) (Li et al, 2016), designed for the long-term propagation of nephron progenitor cells (NPCs), to encapsulate dissociated human fetal kidney tissue in a hydrogel for the generation of human fetal kidney organoids (hFKOs). These self-organizing, 3D epithelial structures, referred to herein as hFKOs, exhibit characteristics of both organoids and organotypic cultures. While they arise from dissociated single cells that self-organize into nephron-like structures, their primary tissue origin and preservation of native developmental hierarchies align them with organotypic systems of tissue slices. hFKOs faithfully recapitulate early developing kidney epithelium, as evidenced by the polarized convoluted 3D structures and expression of key developmental markers at the transcriptional and protein levels. In contrast to adult tubuloids, hFKOs contain complex epithelial structures consisting of developing epithelial cell populations and are not restricted to the tubular compartment of the nephron (i.e., containing podocytes, stroma, and vasculature). Furthermore, as compared to iPSC-KOs, hFKOs show significantly higher expression of early progenitor epithelial genes that are normally expressed mostly in pre-tubular aggregates, RVs, and comma- and S-shaped bodies, such as *PAX2*, *LHX1*, and *JAG1*. Finally, enrichment of Notch pathway-associated gene expression in hFKOs and the critical role of

this pathway in nephron patterning enabled us to study the effects of Notch inhibition at the single-cell level and decipher potential developmental defects in Notch-related congenital kidney anomalies, such as those seen in Alagille syndrome.

# Results

## Establishing a human fetal kidney organoid (hFKO) culture system

Nephron development begins in the first trimester of pregnancy and is completed between weeks 32 and 36 (Rosenblum et al, 2017); therefore, we used whole hFKs from 15–20 weeks' gestation to generate hFKOs. All hFKs were minced, enzymatically digested and dissociated into single cells, embedded in extracellular matrix in the form of basement membrane extract or Matrigel, and cultured in a chemically defined, serum-free medium for multiple (1–8) passages of 2–3 weeks each (Fig. 1A). We tested two different media formulations, hNPSR and AKO, originally designed for human nephron progenitor self-renewal (Li et al, 2016) and adult kidney organoids, respectively, containing components important for the regeneration and homeostasis of adult epithelial tissue, such as RSPO-1 and EGF (Yousef Yengej et al, 2020) (Fig. EV1A). hNPSR supported the growth of complex convoluted and budding epithelial structures (Fig. EV1B), while cells grown in AKO presented a more cystic phenotype (Fig. EV1C), consistent with prior publications showing that adult (Schutgens et al, 2019) and iPSC-derived (Yousef Yengej et al, 2023) kidney cells self-organize into cystic structures dubbed tubuloids. Since the protocols for tubuloid propagation select for mature tubular epithelium, we chose to utilize the hNPSR protocol, as it consistently produced hFKOs strongly resembling early epithelial structures observed in the hFK tissue and is well established for use in preserving nephron progenitor cells (NPCs) (Huang et al, 2024).

With the hNPSR protocol, embedded hFK cells self-organized into convoluted epithelial structures—hFKOs—exhibiting a polarized kidney epithelium, as evidenced by basolateral EPCAM staining and concentrated apical F-actin staining (Fig. 1B) and containing a complex network of lumens (Fig. 1C; Movie EV1). Electron microscopy further demonstrated hFKO polarity, which included luminal microvilli, tight junctions, and even primary cilia (Fig. EV1D). hFKOs contained cells with high abundance of the proliferation marker KI-67 (Fig. EV1E), which enabled their prolonged culture with retention of a convoluted phenotype and expression of key epithelial marker genes such as *CHD1* and *EPCAM* at higher passages, such as P6 (~5 months after dissociation and embedding); their phenotype remained similar to that in the earlier passages P0 (2–3 weeks) and P2 (6–9 weeks) (Figs. 1D and EV1D). To test basic epithelial transport, we added forskolin, a potent activator of cyclic AMP, to the culture medium; this led to water uptake and swelling of hFKOs, indicating a correct orientation of transporters and channels similar to that of native hFK. Interestingly, not all hFKOs swell in response to forskolin (Fig. EV1F). Since CFTR, activated by increased cyclic AMP signaling, is expressed in the proximal and distal tubules (Souza-Menezes et al, 2014) of the hFK, it is likely that early differentiating epithelium in the hFKOs has yet to express CFTR and hence does not respond to forskolin stimulation.

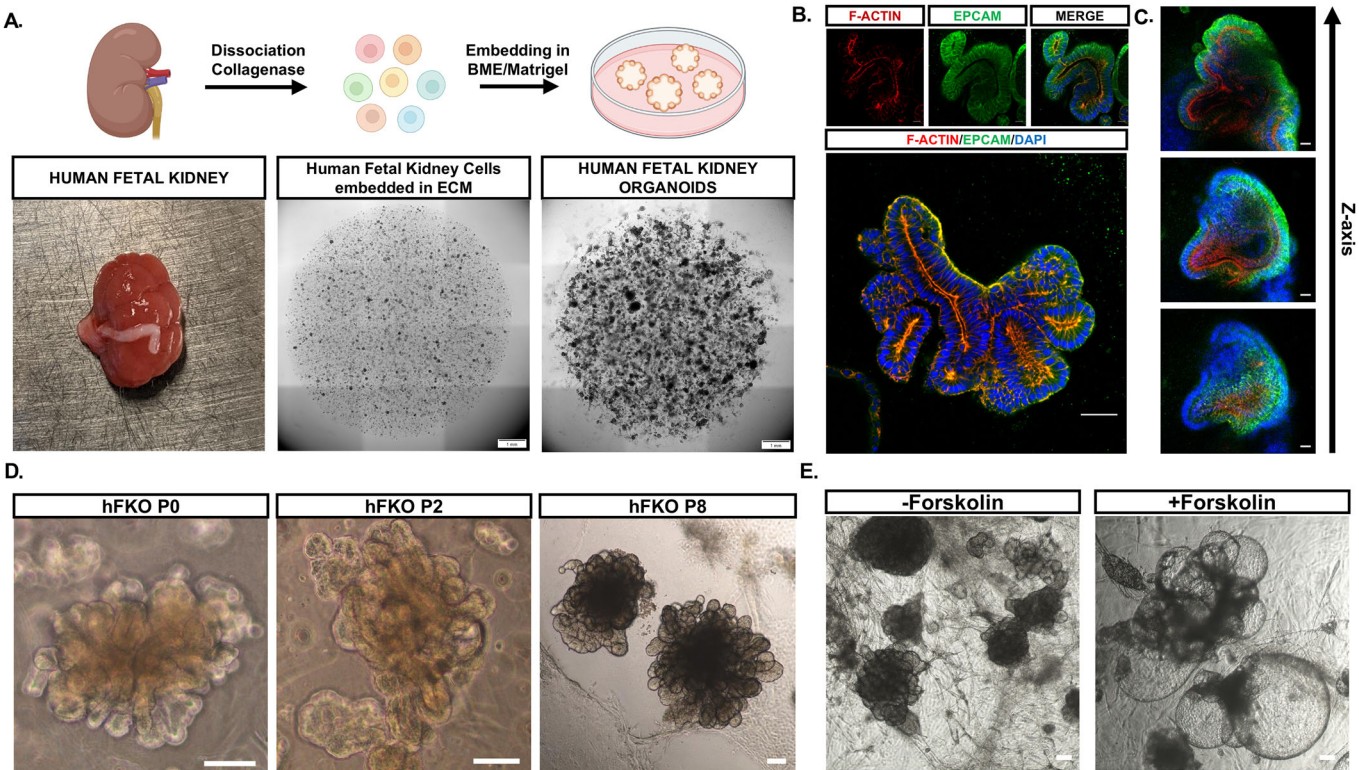

**Figure 1. Establishment of hFKO generation protocol.**

(A) Top: Schematic representation of the process of dissociating, embedding, and culturing hFK cells to form hFKOs. Bottom left: Human fetal kidney before mincing and dissociation. Bottom middle: Widefield image of basement membrane extract (BME) droplets containing hFK cells immediately after dissociation. Bottom right: hFKOs arising after 2–3 weeks of culture. Scale bars, 1 mm. (B) Actin (red, enriched in microvilli) and EPCAM (green, a general epithelium marker) staining reveal a polarized kidney epithelium, similar to that of the native fetal kidney. Scale bars: top, 20 μm; bottom, 50 μm. (C) Intricate luminal networks are detected by confocal microscopy. Scale bars, 50 μm. (D) hFKOs grown in hNPSR for multiple passages, shown at P0 (2–3 weeks after embedding and culture, n = 28), P2 (n = 15), and late hFKOs (P5-P6, P8, n = 3), displaying convoluted epithelium. Scale bars, 100 μm. (E) hFKOs swell in reaction to 25 μM forskolin supplementation to hNPSR medium. Scale bars, 100 μm. Source data are available online for this figure.

## hFKOs strongly resemble the hFK in developmental hierarchy and patterning

We have previously reported a combination of cell surface markers, including NCAM1$^+$ and EPCAM$^-$, that delineates progenitor cell populations in the hFK (Pode-Shakked et al, 2016). Accordingly, we dissociated fresh hFKOs into single cells and performed FACS analysis, which showed that the hFKOs contained early (NCAM$^+$EPCAM$^-$, 17.96%) and more developed (NCAM1$^+$EPCAM$^+$, 20.6%) epithelial progenitors (Fig. EV2A). In P0 hFKOs (freshly dissociated from hFK, embedded in extracellular matrix, and grown for 2–3 weeks), we also observed NCAM1$^+$SIX2$^+$ NPCs, indicating that the stem cells that drive nephrogenesis in the hFK are also present in vitro (Fig. 2A). hFK tissue staining showed that NCAM1 was also found in early developing epithelium, together with PAX2, a key regulator of nephron differentiation that is observed in all epithelial components of the developing nephron (Narlis et al, 2007) as well as the ureteric bud. This early NCAM1$^+$PAX2$^+$RET$^-$ epithelium was also preserved in hFKOs, where it commences its organization into primitive tubular structures—in contrast to NCAM1$^+$SIX2$^+$ NPCs, which have yet to commence differentiation (Fig. 2B). While *NCAM1* expression is downregulated as the nephron matures, *EPCAM*

expression is upregulated; thus, early differentiating NCAM1$^+$ hFKOs, which are widely present in culture, resembled PTAs and RVs in their aggregate structure, whereas EPCAM$^+$PAX2$^+$ hFKOs were tubular and convoluted (Fig. EV2B). Co-expression of vimentin, a marker of cap mesenchyme, with EPCAM in hFKOs indicates ongoing mesenchymal-to-epithelial transition (MET) (Pleniceanu et al, 2010). In addition to epithelial cells, MEIS1$^+$ supporting fetal kidney stroma is also found alongside hFKOs (Naiman et al, 2017) (Fig. EV2C). hFKO stained positive for proximal tubules (Figs. 2C and EV2D), developing WT1$^+$CDH6$^+$ cells, and more mature HNF1B$^+$LTL$^+$ structures, as well as for SYNPO$^+$ podocytes localized next to LTL$^+$ tubules (Fig. EV2E), ECAD$^+$MUC1$^+$ mature distal tubules (Fig. 2D), and ECAD$^+$GATA3$^+$ connecting tubules (Fig. EV2F), confirming the presence of differentiated tubules in the hFKOs alongside the nascent epithelium. Small blood vessels, identified by staining for the endothelial cell marker CD31, could also be detected invading the hFKOs (Fig. EV2G). As the kidney epithelium develops, it undergoes a patterning process governed by distinct, spatially bound transcription factors for each nephron segment, such as LHX1 (distal), JAG1 (medial), and WT1 (proximal) (Costantini and Kopan, 2010). These factors are present in hFKOs, which self-organize into these distinct niches, recapitulating the proximal–distal patterning in vitro (Fig. 2E). Furthermore, CDH6

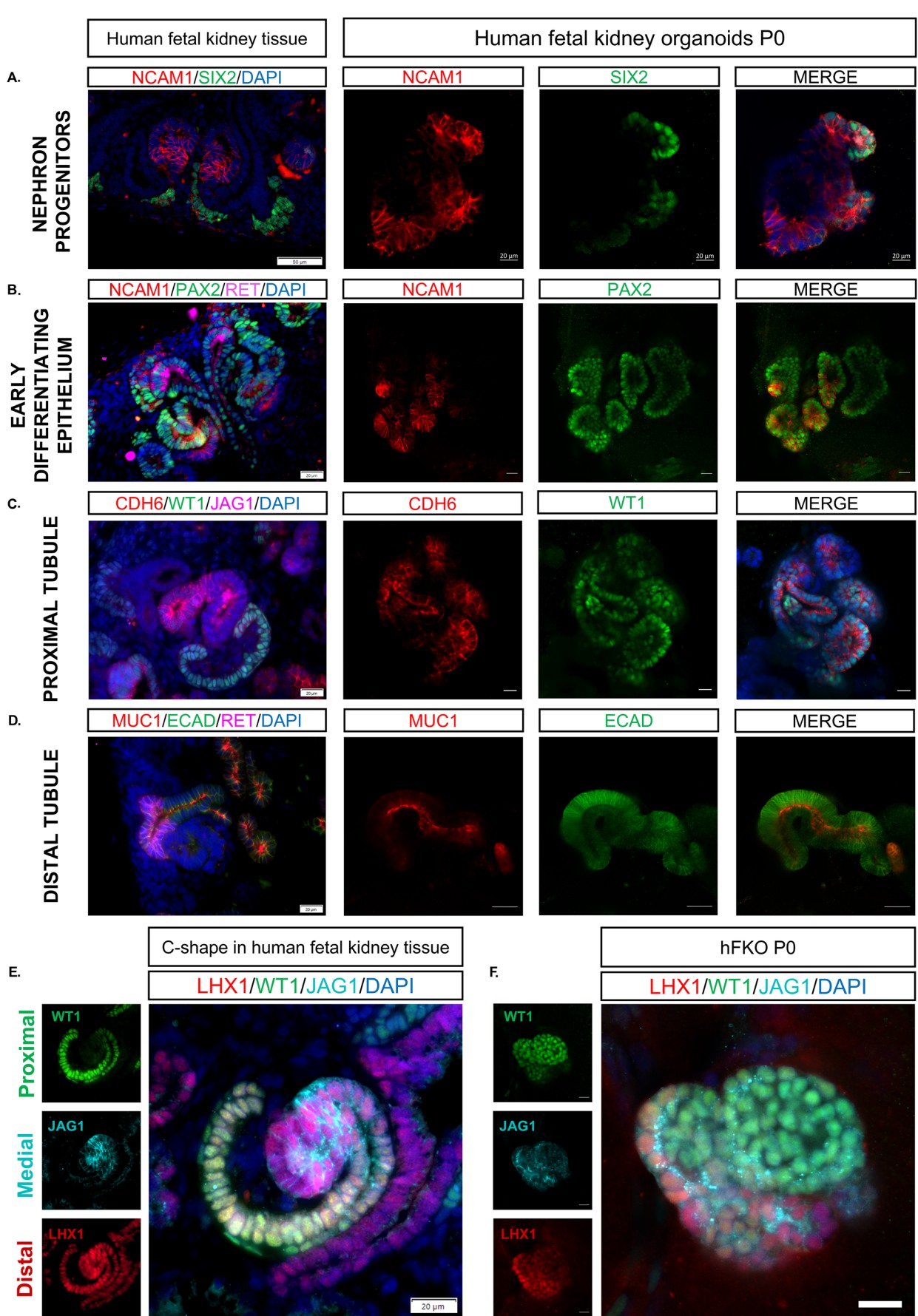

◄ **Figure 2. Developmental hierarchy is preserved in hFKOs.**

Similarly, to the native tissue, hFKOs harbor. (**A**) Nephron progenitor cells which are NCAM1$^+$ SIX2$^+$, scale bars: fetal tissue—50 μm, hFKOs—20 μm. (**B**) Early nephron epithelium which expresses NCAM1$^+$ PAX2$^+$, enabling the growth and proliferation of these organoids. Scale bars: fetal tissue—20 μm, hFKOs—20 μm. (**C**) The proximal tubule has a differentiation axis which involves early WT1 expression, followed by CDH6 and ultimately LTL. These markers are present in human fetal kidney organoids, suggesting that differentiation states are conserved in vitro. scale bars: fetal tissue—20 μm, hFKOs—20 μm. (**D**) In addition to proximal tubules, MUC1$^+$ and ECAD$^+$ distal tubule structures are present in culture. Scale bars: fetal tissue—20 μm, hFKOs—20 μm. (**E**) The C-shape is a hallmark developmental structure in nephron differentiation, in which nephron segmentation begins. Scale bar: 20 μm. (**F**) The segmentation and patterning orchestrated by the proximal WT1, Medial JAG1 and Distal LHX1 markers are recapitulated in hFKOs. Scale bar: 20 μm. Source data are available online for this figure.

and JAG1 staining are correlated, supporting the importance of Notch signaling in the development of proximal tubules (Fig. EV2H). Overall, these staining experiments showed that our hFKOs strongly resemble the developing hFK tissue and reconstitute the developmental hierarchy present in the nephrogenic compartment, encompassing NPCs, early differentiating and mature epithelium, fetal stromal cells, and even small blood vessels.

## Bulk RNA sequencing reveals enhanced expression of developmental genes in hFKOs compared to hPSC-derived kidney organoids

To further characterize hFKOs and to clarify time-dependent developmental processes, we performed bulk RNA sequencing of hFKOs at P0, P2 along with adult kidney organoids (AKOs) and iPSC-derived kidney organoids (iPSC-KOs) for comparison. We initially analyzed hFKOs only and found that they start to initiate differentiation and MET in vitro, as evidenced by the downregulation of key developmental genes such as *CITED1*, *GDNF*, *LHX1*, *JAG1*, *PAX2*, and *DLL1* and upregulation of *CDH6*, *HNF1A*, and *CDH1* (Fig. 3A). P0 hFKOs expressed markers of distinct developmental niches such as PTAs, RVs, CSBs, and SSBs, while P2 hFKOs showed upregulation of markers of mature functional proximal tubules (*SLC4A4, SLC1A1, ANPEP*), distal tubules (*MUC1, CDH16*), and collecting ducts (*AQP3, SCNN1A* and *CDH1*), indicative of a maturation process occurring during hFKO growth (Fig. EV3A). Consequently, as Gene Ontology (GO) enrichment analysis showed, P0 hFKOs are enriched for markers of nephron morphogenesis and renal system development, while P2 hFKOs are enriched for markers of tubular development and morphogenesis (Fig. 3B). Interestingly, markers of both epithelial-to-mesenchymal transition (EMT) and MET are expressed in P2 hFKOs (Fig. EV3B), suggesting that processes of MM proliferation and kidney epithelium differentiation occur simultaneously. Downregulation of the podocyte marker genes *MAFB* and *WT1* between passages may indicate that the dissociation protocol and hFKO passaging can be detrimental for podocyte survival.

hFKOs derived from mid-gestation tissues (week 18) can be maintained in culture for a prolonged period of over half a year. Hence, their growth pattern potentially mimicking aspects of in utero organogenesis can lead over a 5–6-month period to features of a mature kidney (week 40). We therefore performed bulk RNA sequencing of hFKOs at P0 (early) and P5-P6 (late) to further analyze hFKO growth over this in vitro gestational window (Fig. 3C). In line with human kidney development, a shift in expression of developmental genes is seen in late hFKOs, which express general nephron epithelium maturation markers *PAX8* and *CHD1*, proximal (*CUBN*) and distal markers (*MUC1, KRT18*) as

well as a plethora of segment-specific transporters such as *ABBC1, AMN, SLC9A1, AQP3, SCNN1A* and *SCNN1B* (Figs. 3D and EV3C). Therefore, P6 hFKOs are enriched for markers of epithelial cell development and renal absorption, indicating an inverse correlation between the progenitor and mature kidney compartments in timeline paralleling in vivo development (Figs. 3E and EV3D). Importantly, even though we observed a relative downregulation of developmental genes between P0 and P2 hFKOs, we were still able to detect NCAM1$^+$ PAX2$^+$ epithelial progenitors, as well as CDH6$^+$ WT1$^+$ early proximal tubules and SSB-like structures with distal (LHX1$^+$) and proximal (WT1$^+$) segmentation (Fig. 3F). Moreover, to determine the functionality of NCAM1$^+$ PAX2$^+$ epithelial progenitors, we performed magnetic cell sorting (MACS) on P2 hFKOs (Fig. 3G). Prospective isolation of NCAM1$^+$ progenitors from P2 hFKOs demonstrated that hFKOs could reinitiate from NCAM1$^+$ cells but not from negative counterparts (Fig. 3H). Notably, NCAM1$^+$ derived organoids self-organized into epithelial progenitor cell niches expressing NCAM1$^+$ PAX2$^+$, tubular epithelium structures expressing kidney segment markers LTL, MUC1 and HNF1B, and nascent nephron epithelium mostly co-expressing NCAM1 and EPCAM with areas of maturation devoid of NCAM1 (Fig. 3I). We then employed bulk transcriptomics to compare hFKOs to iPSC-KOs. For comparison, we concurrently performed RNA sequencing at sequential differentiation timepoints of iPSC-KOs, specifically days 10 and 18 of differentiation. This approach enabled us to compare hFKOs with various developmental stages of iPSC-KOs, which typically cease growth after 3–4 weeks. Euclidean distance analysis revealed divergent expression patterns which suggest that as hFKO are passaged over time, the similarity between hFKOs and iPSC-KOs decreases (Fig. EV3E). At the start of their culture, P0 hFKOs express higher levels of genes related to distinct developmental niches such as NPCs (*SIX1, SIX2, CITED1, GDNF*), RVs (*LHX1, JAG1, PAX2, DLL1*), and SSBs (*HNF1B, SOX9, POU3F3, CDH6*) (Fig. 3J). hFKOs also express higher levels of mature segment markers such as proximal tubules (*SLC3A1, AQP1, ANPEP*), loop and distal tubules (*SLC12A1, UMOD, SLC12A3*), and collecting ducts (*AQP2* and *CDH1*) (Fig. EV3F). GO term enrichment analysis tool showed that renal system and kidney development genes as well as Notch signaling (Fig. EV3G) were highly enriched in P0 hFKOs (Fig. 3K). Importantly, in comparison to iPSC-KOs, late P5-P6 hFKOs had higher expression levels of epithelial progenitor genes such as *PAX2, PROM1, CD24, HNF1B*, even after 5 months of culture, and are enriched in genes related to epithelial cell development and differentiation (Figs. 3I and EV3H). However, iPSC-KOs had higher expression of podocyte development markers such as *MAFB, OLFM3*, and *WT1*, especially when compared to late hFKOs. Nevertheless, as compared to adult kidney tubules,

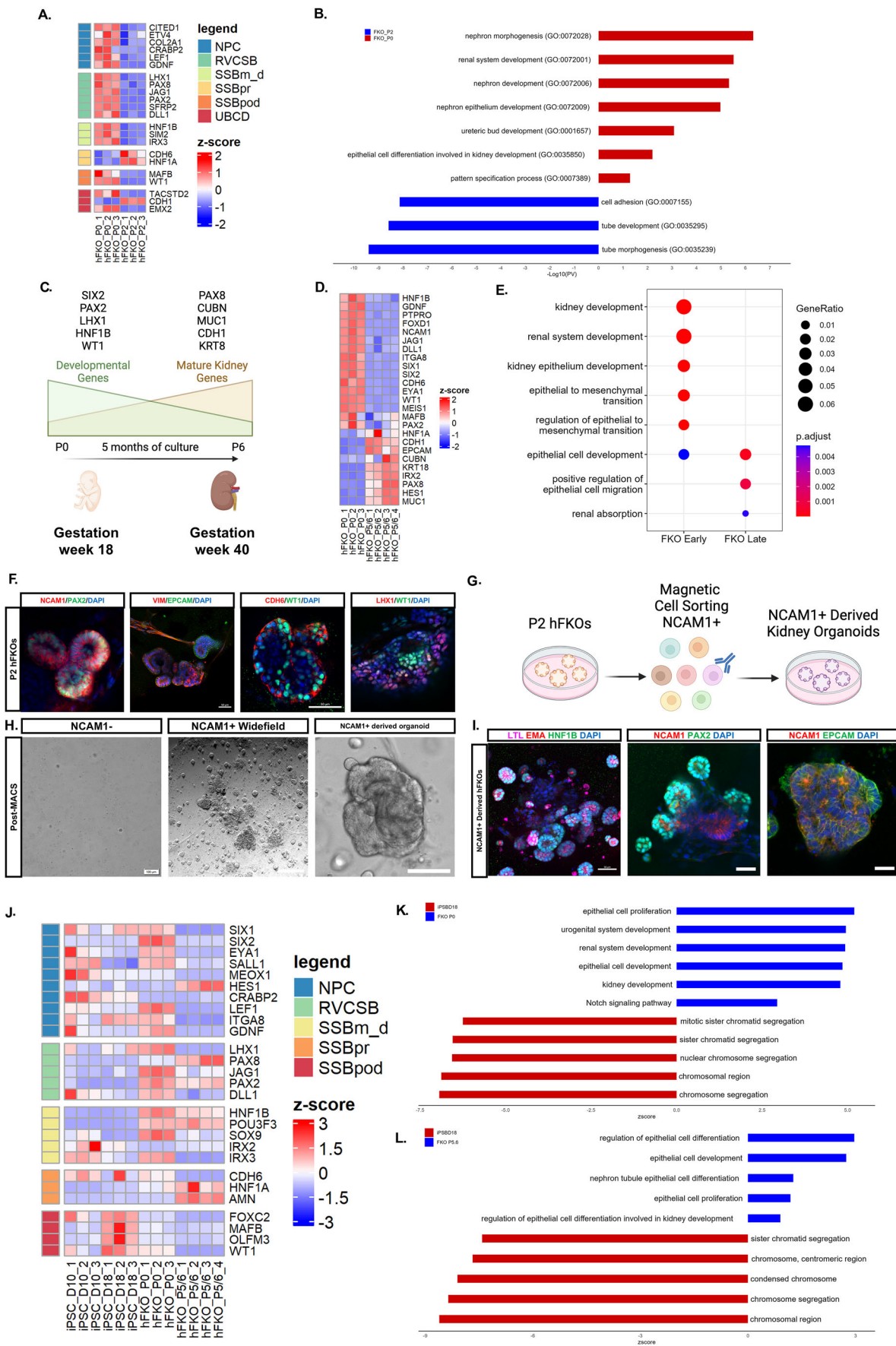

**Figure 3.   Bulk RNA sequencing of hFKOs shows a dynamic development process similar to the native fetal kidney.**

(A) heatmap comparing expression of key NPC, Renal vesicle/comma-shape body (RVCSB), proximal (SSBpr), medial-distal (SSBm_d) and podocyte (SSBpod) compartments of s-shaped bodies (SSBs) and ureteric bud/collecting duct (UBCD) markers in P0 (Freshly dissociated, embedded and cultured for 2–3 weeks) and P2 hFKOs (Organoids which were passaged twice and were cultured for 8 weeks in total). (B) Ontology analysis of P0 (Red) and P2 (Blue) hFKOs, processes of nephron morphogenesis and renal system development are evident in P0, while P2 is characterized by tube development and morphogenesis, in line with expected maturation and differentiation of the hFKO culture. (C) hFKOs cultured from week 18 of gestation for 5 months to p6 (aligning in time with the antepartum kidney, prior to birth), exhibit a maturation process commencing with a high expression of developmental genes such as *SIX2, PAX2, LHX1, HNF1B*, and *WT1* and progressing to a mature tubular kidney epithelium expressing *PAX8* and *CDH1*, as well as segment-specific markers such as *CUBN* (proximal) and *MUC1, KRT8* (distal). While developmental are highly expressed in earlier passages in contrast to mature genes and vice versa, these genes are still expressed in higher passages, hence the trapezoid shape of the expression gradient across time. (D) Heatmap of the developmental program undergoing in hFKOs. P0 hFKOs show high expression levels of developmental genes such as *JAG1, CDH6, WT1, EYA1, NCAM1, SIX2, SIX1*. Late P6 hFKOs exhibit higher levels of maturation markers such as *PAX8, CHD1, EPCAM*, as well as segment-specific markers of proximal (*CUBN*) and distal (*KRT18, IRX2, MUC1*) segments. (E) Ontology analysis of Early (P0) and Late (P6) hFKOs: kidney and renal system development are evident at P0, whereas P6 is characterized by epithelial cell development and renal absorption, in line with the expected maturation and differentiation of the hFKO culture. (F) P2 hFKOs, contain NCAM1+ PAX2+ tubular epithelium progenitors (Scale bar 20um), EPCAM+ VIM+ (Vimentin) tubular epithelium with ongoing differentiation and MET. CDH6+ WT1+ structures containing early proximal tubules and hFKOs with early distal, LHX1+, and proximal, WT1+, compartments of the SSB. Scale bars, 50 µm. (G) Scheme of isolation of NCAM1+ cells, by magnetic cell sorting (MACS) from P2 hFKOs, reveal the ability of these cells to form organoids (H) Brightfield images of NCAM1-negative fraction, devoid of newly grown hFKOs (left, scale bar 100um), Widefield NCAM1-positive fraction, giving rise to hFKOs (middle, scale bar 200 µm, n = 4), Zoomed image of a NCAM1+ derived kidney organoids (right, scale bar 100 µm). (I) IF of NCAM1+-derived kidney organoids containing LTL+ MUC1+ HNF1B+ proximal–distal tubular epithelium (left, scale bar 50 µm), additional NCAM1+ PAX2+ epithelial progenitors (Middle, scale bar 20 µm) and EPCAM+ NCAM1+ nascent nephron epithelium (right, scale bar 20 µm). (J) Heatmap of developmental genes, comparing expression between iPSC-KOs at early (D10) and late (D18) stages of differentiation, to hFKOs at P0 and P6. (K) Ontology analysis of Early hFKOs (P0, blue) vs differentiated iPSC-KOs (iPSCBD18, red): enrichment of genes correlating to urogenital and renal system development as well as epithelial cell proliferation and kidney development, while mitosis-related processes are evident in iPSC-KOs. (L) Ontology analysis of Late hFKOs (P5 and P6, blue) vs differentiated iPSC-KOs (iPSCBD18, Red): After 5 months of culture, P6 hFKOs are enriched with genes governing epithelial cell development as well as metanephric nephron epithelium development. Similar to the comparison to early hFKOs, iPSC-KOs are enriched with proliferation-related genes that regulate cell division and growth. Source data are available online for this figure.

which are solely tubular in nature, hFKOs showed high expression of podocyte markers such as *NPHS1, NPHS2*, and *PODXL*, further supporting the existence of a podocyte niche in hFKOs (Fig. EV3I). Furthermore, markers of off-target cell populations showed lower expression in hFKOs than in iPSC-KOs, as exemplified by *CNTNAP2 and ZIC2* (neurons) and *MYL1*, and *TNNC2* (muscle) expression levels (Fig. EV3J).

Finally, further analysis and comparison with publicly available bulk RNA data from common hPSC-KO protocols (Gupta et al, 2022; Takasato et al, 2015) and developmental timepoints revealed that expression of *Notch1, Notch3, LHX1, DLL1*, and *HES1* was significantly higher (up to 10-fold) in P0 hFKOs than in their hPSC-KO counterparts, while *JAG1* expression levels were on par with those of hPSC-KOs (Fig. EV3K). In the context of Notch signaling, only *Notch2* and *HEY1* expressions were lower. However, the key epithelial progenitor markers *CD24, HNF1B, IRX3*, and *PAX2* showed considerably higher expression in hFKOs (up to 100-fold; Fig. EV3L). Expression of *PAX2* was lower at all timepoints of hPSC-KO differentiation with comparative staining showing a reduction in co-expression of NCAM1 and PAX2 as the PSC-organoids differentiate, observed already at day 18 in hESC-KO and iPSC-KOs (Fig. EV4A,B) while NCAM1+ PAX2+ structures could still be detected at P2 hFKOs (Fig. EV4C). Concomitantly, nephron epithelial markers *EPCAM, CDH16, MUC1*, and *KRT19*, expression was increased at P2 hFKOs, indicating a differentiation and maturation process not recapitulated in hPSC-KO counterpart (Fig. EV3M). These results indicate that hFKOs are a suitable model for studying Notch pathway-related CAKUTs, establishing the hFKOs as a dynamic system in which early epithelial progenitors differentiate and mature, expressing high levels of key evelopmental signaling pathway genes and encompassing the plethora of nascent and mature niches present in the evolving hFK.

## Single-cell RNA sequencing and pseudotime analysis show hFKOs recapitulate the native hFK developmental hierarchy in vitro

To fully characterize the highly heterogeneous hFKO culture system, we utilized single-cell RNA sequencing (scRNA-seq) to further elucidate the distinct cell niches and developmental processes in hFKOs. Two healthy hFKs (from weeks 15 and 20 of gestation) were cultured for 3 weeks, sampled separately, and pooled together to form an hFKO single-cell dataset containing 13,368 cells (after QC filtration). UMAP clustering revealed 15 clusters, of which 10 are nephrogenic, consisting of genes with differential expression: NPCs (*CRABP2, LYPD1*), RVs (*PAX2, PAX8, LHX1*), CSBs (*CD24, IGFBP7, PAX2, CDH6*), early podocytes (*OLFM3, MAFB*), mature podocytes (*NPHS2, PTPRO, PODXL*), early distal tubule (*POU3F3, CLDN3*), and mature proximal (*FXYD2, LRP2*) and distal convoluted and connecting (*MUC1, AQP2*) tubules (Fig. 4A,B). The clustering also detected a ureteric bud–derived collecting duct cluster (*KRT17, KRT19*) and non-epithelial clusters such as an interstitial cluster (*COL3A1, COL1A1*), an endothelial cluster (*PECAM1*), and a small cluster of "stressed" cell expressing cell-stress markers (*MALAT1* (Hirata et al, 2015), *NEAT1* (Liu et al, 2017)). Notably, when non-epithelial clusters were filtered out, both hFK samples contained the same nephron-associated clusters, with little variability in cell number (Fig. EV5A). One sample (from week 20 of gestation) was derived from thawed dissociated hFK tissue; since both fresh and frozen samples produced the same clusters, hFK samples can be stored long term for the generation of hFKOs without detrimental effects. hFKOs can also be successfully frozen for long-term storage and thawed for later experiments (Fig. EV5B). The hFK samples contribute different amounts of cells to the two podocyte clusters (Pod-1 and Pod-2) expressing *NPHS1, NPHS2*, and *PODXL*.

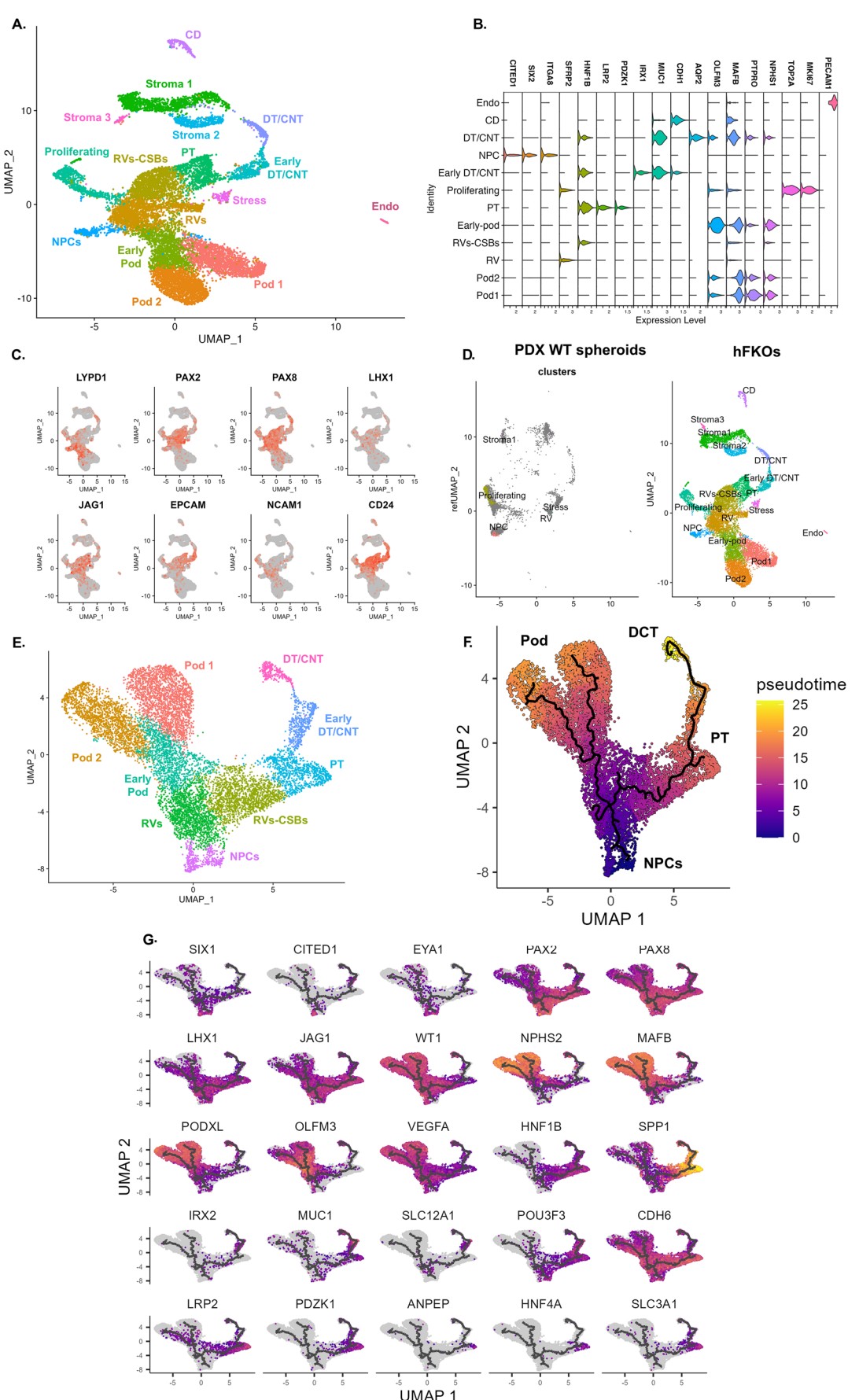

**Figure 4.   Single-cell RNA sequencing of hFKOs reveals developmental processes similar to those in the native fetal kidney.**

(A) UMAP clustering of P0 hFKOs reveal 16 clusters: nephron progenitors (NPCs), renal vesicles (RVs), renal vesicles and comma-shaped bodies (RVs-CSBs), early podocytes (Early Pod), podocytes (Pod-1, Pod-2), proximal tubules (PT), early distal tubule/connecting tubules (Early DT/CNT), distal/connecting tubules (DT/CNT), *TOP2A*+*KI-67*+ cells (Proliferating), stressed cells (Stress), stromal cells (Stroma 1, Stroma 2, Stroma 3), collecting ducts (CD), and endothelial cells (Endo). (B) Violin plots of the expression of key markers used to characterize clusters. (C) Feature plots of markers depicting development and regeneration states, NPCs (*LYPD1*, *PAX2*, NCAM1), differentiating epithelium (*PAX8*, *LHX1*, *JAG1*) differentiated epithelium (*EPCAM*), and cells with regenerative capacity (*CD24*). (D) Projection of hFKOs on PDX-Wilms tumors, NPCs, and proliferating cells from hFKOs match blastema tumor cells. (E) UMAP clustering of the nephrogenic compartment of hFKOs after sub-setting. (F) Pseudotime plot of nephrogenic compartment, starting at NPCs and bifurcating into podocytes and tubular lineages. (G) UMAP clustering of developmental markers across pseudotime axis in nephrogenic compartment of hFKOs. Source data are available online for this figure.

Differential expression analysis between clusters Pod-1 and Pod-2 reveals higher expression of PTPRO (Tran et al, 2019), a mature podocyte marker in Pod-1. Since ~60% of the cells in cluster Pod-2 originated from a week 15 hFK, a larger proportion of podocytes were still developing than among the podocytes from a week 20 hFK. The difference may derive from the gestational age of the original hFKs, as older hFKs encompass more mature cell populations. Spanning the NPC-RV-CSB-proximal/distal tubule differentiation axis are key developmental and stemness genes including *PAX2*, *PAX8*, *LHX1*, *JAG1*, *EPCAM*, *NCAM1*, and *CD24* (Fig. 4C). Interestingly, a proliferating cluster, defined by high expression of *TOP2A* and *MKI67*, also expresses *PAX2*, *PAX8*, and *NCAM1*, a possible driver of hFKO propagation. To further exemplify the diversity of hFKO, we generated an scRNA-seq dataset from a patient-derived Wilms tumor (WT) xenograft that was enriched for the WT blastema (Pode-Shakked et al, 2013), the tumor compartment containing the NPC-like malignant cells in WT. When the hFKO scRNA-seq dataset was projected upon the blastema-enriched WT counterpart, only the NPC and proliferating clusters from the hFKO dataset aligned with the WT blastema, which is highly enriched for SIX2+ cells (Markovsky et al, 2017; Murphy et al, 2019) (Fig. 4D). In addition, when the hFKO scRNA-seq dataset was integrated with week 17 hFK cortex-derived scRNA-seq dataset (Lindström et al, 2018b, 2018a), most cortex-derived hFK cells aligned with NPCs, proliferating cluster (TOP2A+KI-67+), and RVs, whereas more differentiated cell clusters, such as CSBs, podocytes, and distal and proximal tubules, were less represented (Fig. EV5C). Thus, comparative scRNA-seq illustrates wide cell population diversity of the hFKO cultures in comparison with samples that highlight a more specific developmental stage. Pseudotime analysis is a robust tool for visualizing differentiation states in a heterogenous tissue or culture, which showed that hFKOs preserve the differentiation trajectory observed in previous scRNA-seq works on human (Wang et al, 2018a; Lindström et al, 2018a) and mouse (Combes et al, 2019; Miao et al, 2021) fetal kidneys. For example, NPCs in hFKOs bifurcate into two lineages at the RV stage, leading to podocytes and proximal–distal early tubules, similar to other observations of the native hFK (Fig. 4E,F). The pseudotime heatmap showed NPCs and early epithelium (expressing *SIX2*, *PAX2*, *JAG1*, *LHX1*, and *NCAM1*) at the root of the pseudotime tree, which bifurcates into a differentiated tubular epithelium (*LRP2*, *EPCAM*, *CDH6*, *PAX8*, *HNF1B*) and developing and mature podocytes (*WT1*, *MAFB*, *NPHS1*, *NPHS2*, *PODXL*) (Fig. 4G). Interestingly, *VEGFA* expression increases as the podocytes develop in the hFKOs, correlating to the generation of vascular clefts, which in the developing hFK induces migration of blood vessels into the newly formed glomeruli (Eremina and Quaggin, 2004). We wished to compare our hFKO

scRNA-seq data to other publicly available datasets from hFK tissue and PSC-derived kidney organoids. We utilized a recently developed tool, DevKidCC, that facilitates cell classification and comparison between various cell sources and protocols of in vitro kidney models (Wilson et al, 2022). DevKidCC identified and classified the various nephrogenic clusters in an unbiased manner, with results that matched our original classification via cross-referencing known markers in the literature (Fig. EV5D). When compared to the dataset included in the DevKidCC tool, which comprises PSC-derived kidney organoids as well as hFK tissue references, hFKOs at P0 were highly enriched with early developing nephron cells ("EN", *LHX1*, *JAG1*), early podocytes ("Epod"), and a committed NPC-like population similar in proportion to the hFK reference (Fig. EV5E). These results confirmed the retention of developmental hierarchy in hFKOs, with a preserved differentiation axis as seen on pseudotime analysis and as compared to hFK tissue and hPSC-derived kidney organoid models.

## Progenitor and mature proximal/distal cell fates are Notch dependent in hFKOs

Our finding that hFKOs have highly enriched expression of Notch signaling pathway genes, on par with that observed in their hPSC-derived counterparts, afforded the opportunity to study the pathway's role in developing human kidney. This analysis could provide insight into the malformed kidneys that appear in Alagille syndrome arising from mutation in Notch receptors and ligands (Kamath et al, 2013). The Notch signaling pathway is known to be crucial to nephron development, with Notch contributing to nephron segmentation and its inhibition or loss leading to ablation of proximal tubules and podocytes (Kopan et al, 2007). Recent work showed that Notch also contributes to distal tubule derivation, thus driving the proper development and maturation of the whole nephron (Chung et al, 2017). A revisiting of earlier work utilizing NCSTN-deficient iPSC-KOs further fortified the evidence for Notch's role as a crucial driver of proximal tubule development (Duvall et al, 2022). When cultured in the presence of the γ-secretase inhibitor DAPT, a potent Notch inhibitor, hFKOs show diminished propagation and form complex, convoluted epithelial structures (Fig. 5A). Notch inhibition sustains NPCs in their SIX2+ state (Yuri et al, 2015; Tanigawa et al, 2016) and hinders their differentiation past the PTA state (Cheng et al, 2003). In accordance with these observations, our reverse transcription–qualitative PCR (RT-qPCR) analysis of inhibited hFKOs showed downregulation of developmental genes such as *LHX1* and *JAG1* as well as downstream target genes such as *HEY1*, while the NPC marker *SIX2* was upregulated; also upregulated were the mature proximal tubule and Loop of Henle (Kortenoeven and Fenton, 2014) marker *AQP1* and the distal tubule marker *MUC1* (Fig. 5B). Immunostaining of treated hFKOs for the proximal markers CDH6 and LTL showed significant damage to proximal

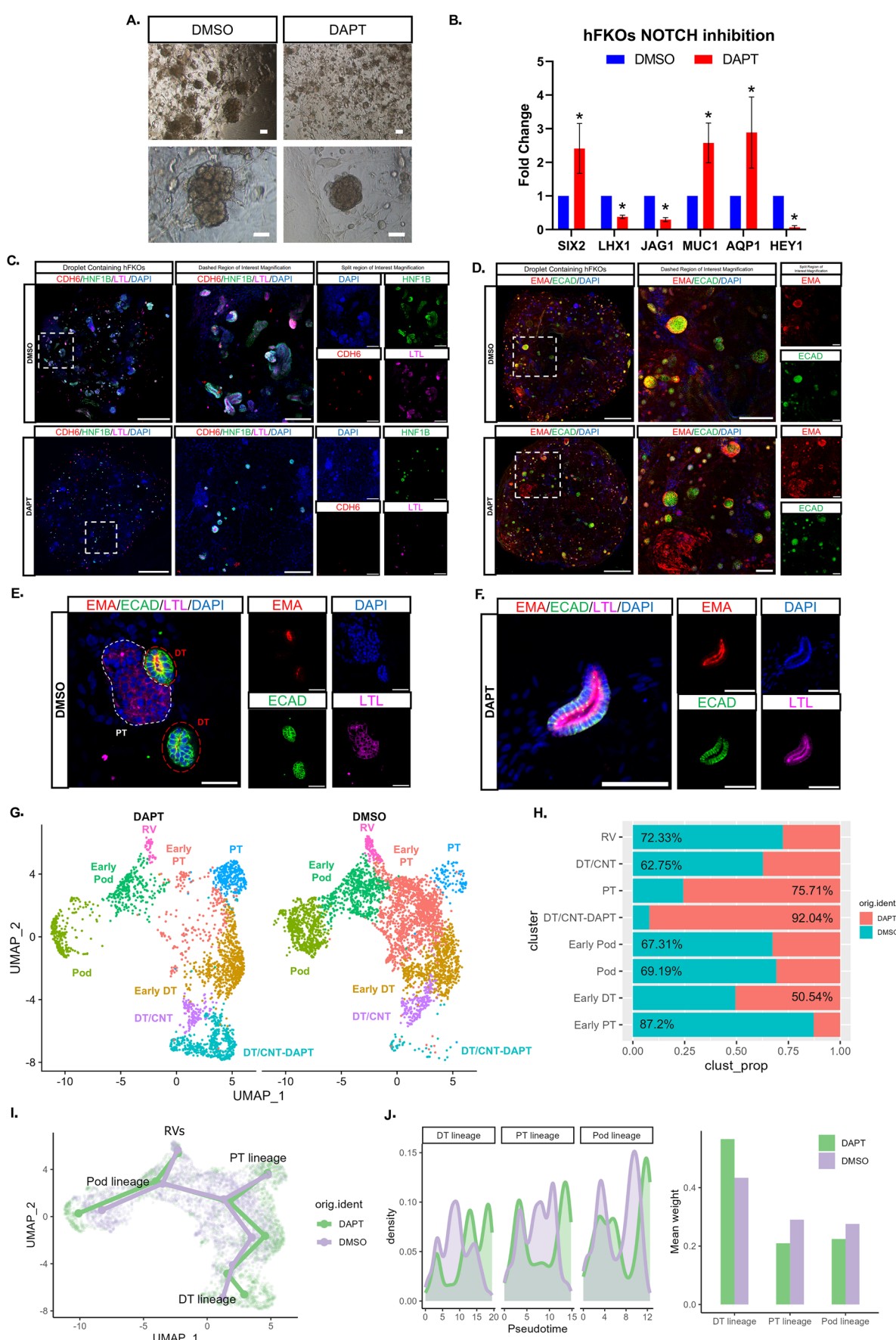

**Figure 5.  Proximal tubule lineage is hampered under Notch inhibition in hFKOs.**

(A) Brightfield images demonstrating the effect of DAPT on hFKOs—a decrease in convoluted structures, proliferation, and an increase in compact distal structures. Top panels: low magnification; bottom panels: high magnification of the same field. Scale bars, 100 μm. (B) Student's *t* tests were used to assess significance in qRT-PCR analysis of NPC (SIX2), developing epithelium (LHX1, JAG1), mature proximal (AQP1), and distal (MUC1) markers, as well as the Notch downstream transcription factor gene HEY1. $n = 3$. Expression levels in untreated hFKOs were used for normalization. Data are presented as mean ± SD. *$P < 0.05$. (C) Upon immunostaining, cells expressing markers of developing (CDH6+HNF1B+) and mature proximal tubules (LTL+) are sparser after treatment with the Notch inhibitor DAPT than after control treatment (DMSO). Scale bars, Droplet—1000 μm, magnification—200 μm. (D) Distal markers EMA and ECAD are present to similar degrees after DAPT treatment and control treatment. Scale bars, Droplet—500 μm, magnification—DMSO 200 μm, DAPT 100 μm. (E) Control-treated hFKOs exhibit distinct niches of proximal (LTL+, white dashed line) and distal tubules (EMA+ECAD+, dashed red line). Scale bar, 100 μm. (F) In contrast, among DAPT-treated hFKOs, triple-positive tubules are more common, potentially indicating incomplete/aberrant segmentation. Scale bar, 100 μm. (G) scRNA-seq of hFKOs under Notch inhibition (DAPT) and control (DMSO), with UMAP of the nephrogenic compartment. RV renal vesicle, Early PT early proximal tubule, PT proximal tubule, Early Pod early podocyte, Pod podocyte, Early DT early distal tubule, DT/CNT distal tubule/connecting tubule, DT/CNT-DAPT distal tubule/connecting tubule enriched under Notch inhibition. (H) Proportions of cells in each cluster in control and treated hFKOs. (I) Pseudotime plot of the three main lineages: podocyte lineage (Pod), proximal tubule lineage (PT), and distal tubule lineage (DT). (J) Cell density vs. pseudotime across different condition: the DT lineage shifts right under DAPT treatment, indicating an enhanced maturation process, while a gap forms in the PT lineage, demonstrating the deleterious effect of Notch inhibition on proximal tubule differentiation. Source data are available online for this figure.

tubule structures, which were withered and shrunken compared to the control (Fig. 5C). In contrast, structures expressing the distal markers EMA and ECAD were less affected, exhibiting similar structures to controls (Fig. 5D). hFKOs usually represent either a part of the nephron, such as adult tubuloids, or the whole nephron, consisting of spatially distinct EMA+ECAD+ distal segments and LTL+ proximal segments, reminiscent of hFK tissue (Fig. 5E). However, under Notch inhibition by DAPT, we detect increased amounts of unusual structures that are triple positive, EMA+ECAD+LTL+, suggesting that Notch inhibition can increase the production of "confused" hFKOs that are abnormally segmented (Fig. 5F). To further characterize the effect of Notch inhibition, we performed scRNA-seq on DAPT-treated and untreated hFKOs derived from week 20 hFK. We detected 16 clusters in the merged dataset, consisting of NPCs, early developing epithelium, and differentiated tubules (proximal and distal), podocytes, stromal cells, and endothelial cells (Fig. EV6A). Larger proportions of NPCs (Fig. EV6B,C) and proliferating cells (expressing *SIX2, PAX2, TOP2A*) (Fig. EV6D) were found in hFKOs under DAPT treatment. Strikingly, the enhanced NPC numbers that emerged in hFKOs under DAPT treatment, are also annotated by DevKidCC as "true" NPCs, which are exclusive to the reference native hFK and appear to lesser extent in hFKOs or PSC-KOs (Fig. EV6E). To investigate the segmentation process of distal and proximal tubules, we focused on the nephrogenic subset, starting with RVs and trifurcating into podocytes and proximal and distal tubules (Fig. 5G). Early proximal clusters expressing early markers such as *CDH6* (Fig. EV7) were reduced in number (12.8% DAPT vs. 87.2% DMSO, Fig. 5H), while differentiated *LRP2*- and *SLC3A1*-expressing mature proximal cells were unhindered and even constituted a higher proportion of cells in the proximal tubule cluster than control. Interestingly, in DAPT-treated hFKOs, an additional cluster of distal tubules was detected with enhanced maturation, expressing *MUC1* and *CDH1*; we dubbed this cluster "DT/CNT-DAPT". In contrast to the proximal lineage, similar proportions of cells expressing *POU3F3* and *TFAP2B* (the early distal cluster) were found in the control and treated groups, suggesting that the distal lineage is undisturbed by Notch inhibition. To further visualize the differentiation trajectories in the two sets of conditions, we utilized the condiments(Roux de Bézieux et al, 2024) workflow (Fig. 5I). An imbalance plot highlights the cell population mismatch in the early proximal cluster between the DAPT and DMSO treatment groups (Fig. EV8A).

This cluster bifurcates into podocyte and tubular lineages, after which the tubular lineage further divides into proximal and distal lineages. Comparison of the pseudotime cell density of each lineage across the two conditions demonstrates the shift of differentiation

states caused by Notch inhibition. The distal (DT) lineage is more mature in the DAPT group, as evidenced by the increased cell density at higher values on the pseudotime axis (Fig. 5J), while the proximal (PT) lineage displays a distinct gap between early and mature proximal tubules, representing the arrest of developing proximal tubules and escape of mature proximal tubules. To explain the different effects of Notch inhibition on the segmentation of distal and proximal tubules, we utilized monocle to subdivide the previously established trajectories into modules that represent different gene patterns across the pseudotime axis (Fig. EV8B). One differentially expressed gene, *MECOM*, showed an expression pattern that bypasses the early proximal tubule cluster, with its expression beginning to increase at the early distal tubule stage (Fig. EV8C). In a zebrafish model, *MECOM* promotes distal segmentation upstream of Notch (Li et al, 2014), thus providing a possible explanation for the development of the distal lineage despite the inhibition of Notch.

In concordance with the abnormal double-positive (LTL+MUC+) hFKOs detected under Notch inhibition, we observed that the DT/CNT-DAPT cluster encapsulates both PT and DT lineages. Isolation and re-clustering showed that the DT/CNT-DAPT is further subdivided into four clusters: Bipotent progenitors, early DT/CNT, mature DT/CNT, and PT (Fig. EV9A). The bipotent progenitor niche expresses early proximal markers such as *CDH6* and *PDZK1* and early distal markers *POU3F3, IRX3*, and *TFAP2B* (Fig. EV9B). The PT cluster expresses early and mature markers such as *CDH6, PDZK1, LRP2*, and *SLC3A1*, whereas the early and mature distal clusters are mainly distinguished by downregulation of *TFAP2B* and upregulation of *MUC1* and *CDH1*. Interestingly, the DT/CNT-DAPT cluster differentially expresses *PROM1 (CD133)*, a marker of clonal cell proliferation in the adult (Osnat et al, 2020; Bussolati et al, 2005) and fetal kidney (Pode-Shakked et al, 2016, 2017) that was recently associated with an adaptive state following kidney injury (Lake et al, 2023) (Fig. EV9C). High expression levels of *ALDH1A1*, encoding a retinaldehyde dehydrogenase that is part of the retinoic acid pathway, can be detected in the PT cluster of DT/CNT-DAPT. Furthermore, the bifurcation of proximal and distal lineages in the DT/CNT-DAPT is evident from the distribution of key markers of early PT, mature PT, early DT, and mature DT in each branch, overlapping in the bipotent progenitor cluster in which cells express early PT and DT markers (Fig. EV9D). Finally, we detected rare "confused" cells in the DT/CNT-DAPT population that

co-express *SLC3A1* and *MUC1* (Fig. EV9E); these were 2.4 times more common under Notch inhibition (Fig. EV9F). Our findings support the notion that proximal tubule segmentation is mainly driven by Notch signaling, as demonstrated by a diminished early proximal niche in DAPT-treated hFKOs.

# Discussion

The hFK is an intricate organ that requires complex spatial and temporal cues for correct development. Most ex vivo 3D models of hFK arise from PSC differentiation to kidney organoids. Herein, by using a collection of mid-gestation hFKs and defined media, we have introduced an in vitro model of the hFK, containing at its onset a large population of NCAM1+ progenitors and incorporating the developing hierarchy of the native hFK, from NPCs to developing and mature nephron tubular epithelium, such as proximal and distal tubules as well as podocytes. The polarized kidney epithelium closely resembles the native FK in its 3D architecture and function; in addition, spatial segmentation into proximal, medial, and distal compartments is evidenced by distinct niches controlled by key developmental factors such as LHX1, JAG1, and WT1. Moreover, hFKOs are not confined to the epithelial lineage but also harbor delicate fetal renal stroma. We went on to show, via bulk and single-cell RNA sequencing, the diverse populations of cells residing in hFKOs. Moreover, comparative analysis of hFKOs and hPSC-derived kidney organoids demonstrated significant enrichment of epithelial progenitors and early epithelial differentiation in the former, allowing the study of developmental processes and diseases in a dish associated with this developmental window of opportunity. In fact, to our knowledge, there is no straightforward in vitro system that mimics these stages of human kidney development as hFKOs do. Accordingly, expression of genes related to critical signaling pathways such as Notch is highly enriched in hFKOs. Previously, using mouse models, Notch has been implicated in proximal segmentation (Duvall et al, 2022; Kopan et al, 2007) in the developing kidney, while additional studies suggested that developing epithelia of both proximal and distal origin are hampered by Notch loss (Chung et al, 2017).

Here, we utilized a single-cell RNA sequencing approach to delineate affected cell sub-populations following pharmacological Notch inhibition of hFKOs. Our data showed that while developing proximal tubules are hindered, the developing distal lineage mostly escapes destruction and the mature distal cluster becomes more prominent, suggesting not only distal escape but rather enhanced distal maturation at the expense of the proximal counterpart. Thus, while hFKO normal growth is clearly Notch dependent, our system supports previous findings of mainly proximal targeting and unravels enhanced distal maturation as a new phenomenon following Notch inhibition. One possible explanation for the selective ablation of early proximal tubules in hFKOs under Notch inhibition is the activity of MECOM, a transcription factor that promotes distal tubule fate in zebrafish, opposing retinoic acid (RA) activity that promotes proximal segment development (Morales et al, 2018; Marra et al, 2019; Li et al, 2014). According to our findings, *MECOM* expression begins in the RV and continues into the early and mature distal tubules. Interestingly, under Notch inhibition, PROM1 is upregulated in the enhanced distal cluster (DT/CNT-DAPT), which incorporates cell niches with combined distal and proximal lineages. The upregulation of ALDH1A1 in the proximal niche and MECOM at the distal niche mimics the segmentation process observed in zebrafish—RA signaling promotes proximal tubule differentiation, while MECOM supports distal tubule formation. The fact that

PROM1 has been previously associated with an adaptive response to kidney injury in the proximal and ascending loop of Henle, and the existence of the physiological RA-MECOM proximal–distal patterning under its influence, suggest that PROM1 could be a marker of a compensatory reaction to malformed proximal tubules and podocytes that arise under Notch inhibition. Even though mouse and iPSC-KO models of Notch inhibition have shown the disruption of proper proximal tubule maturation, we detected AQP1+ and LRP2+ cells in qRT-PCR and scRNA-seq experiments, respectively. Since hFKOs are derived from a heterogeneous mixture of cells, it is possible that AQP1+ and LRP2+ cells were already present in culture or had already set in their fate and thus were not hindered by Notch inhibition. Another possible explanation could be that some cells have escaped the transient pharmacological inhibition by DAPT; however, the significant reduction in cells undergoing proximal tubule development in the early PT cluster and the rise of a newly adaptive CD133+ cluster imply that the inhibitory effect of DAPT is sufficient to study the deleterious effects of Notch mutations on normal hFK development. Of note, 75% of Alagille syndrome-causing mutations in *JAG1* are the result of frameshift, nonsense, or splice-site alterations, suggesting that a possible mechanism of the disease is *JAG1* haploinsufficiency (Crosnier et al, 1999; Spinner et al, 2001), achieved in vitro by pharmacological inhibition of Notch. Even though patients with Alagille syndrome have mutations in *Notch*, which have been shown in mouse and PSC-based models to disrupt proper proximal tubule and podocyte differentiation (Costantini and Kopan, 2010; Kopan et al, 2007; Duvall et al, 2022; Cheng et al, 2003), histopathological imaging reveals that sclerotic podocytes and defected immature proximal tubules intermingled with healthy counterparts can still be detected in samples from such patients (Davis et al, 2010; Bissonnette et al, 2017). The renal manifestations of Alagille syndrome, such as kidney dysplasia and distal tubular acidosis, are mild in comparison to the hepatobiliary disease (Jacquet et al, 2007), which is usually a more prominent manifestation of the syndrome, and the existence of a compensatory mechanism could explain this less severe renal phenotype. The effectiveness of a compensatory adaptive mechanism as well as the undesired derivation of double-positive (expressing distal and proximal tubule markers), "confused", maladaptive cells could determine the severity of the renal manifestation in Alagille patients.

Current limitations of the PSC-derived kidney organoid technology include the availability of only short-term culture methods for the developing part of the hFK, i.e., for NPCs and epithelial progenitors arising from NPCs through the consecutive developmental stages of pre-tubular aggregates, renal vesicles, and comma- and S-shaped bodies. Thus, any extension of the culture period is desirable. Our transcriptomic comparison of propagated hFKO cultures to hPSC-derived kidney organoids derived using multiple protocols and analyzed at varying timepoints along the hPSC-KO growth trajectory showed the retention of epithelial progenitors in propagated hFKO, which was verified by immunofluorescence, and the persistent significant enrichment of genes related to kidney epithelial progenitors and hence developmental stages. Moreover, the prospective isolation of NCAM1+ developmental progenitors in hFKOs at P2 and the ability to further initiate hFKOs from the NCAM1+ fraction support the presence of a degree of ongoing nephrogenesis in a time frame that has no parallel in the literature. In fact, both progenitor-, maturing and segment-specific kidney structures were established in these hFKOs. Although these findings suggest a degree of progenitor renewal and differentiation, the challenge of lineage tracing in primary cultures limits our ability to definitively conclude that these structures arise solely through directed differentiation rather than selective

expansion in culture of existing mature kidney epithelium. In any scenario, this potential for "developmental" expansion may still be harnessed to increase the available biomass for downstream analysis as well as for studying the effects of various pathways on the maturation of kidney tubules over prolonged culture. For instance, our finding that treatment of hFKOs with DAPT increases the number of NPCs and shifts their identity towards a bona-fide NPC state, could potentially introduce a new method for human NPCs propagation based on a maturational block, similar to recently reported methods based on hPSC (Huang et al, 2024). Finally, our bulk RNA sequencing indicated that hFKOs, in addition to showing preservation of the developing epithelium, also breach a certain maturation barrier that hPSC-KOs do not cross, with higher expression of CDH16, EPCAM, EMA, and KRT19 at almost all the timepoints. Thus, both developing and mature epithelial compartments co-exist in hFKO at levels not described in hPSC-KOs.

Our study had multiple limitations; even though we show evidence of ureteric bud and CD cells from single-cell and bulk RNA sequencing, we failed to achieve valid staining of RET[+] or AQP2[+] cells, potentially restricting hFKO diversity. In addition, our use of difficult-to-culture primary cells may limit the capacity to perform genetic manipulation of these cultures. In this regard, the establishment of human fetal hepatocyte organoids (Hendriks et al, 2021) genetically engineered using CRISPR-Cas9 editing warrants cautious optimism about the feasibility of developing similar techniques in hFKOs with considerable efficiency. Finally, a potential disadvantage of our method is that we dissociate and culture single cells from the entire human fetal kidney in a model that encompasses a plethora of niches, mainly nephrogenic and stromal and to a lesser extent, ureteric and endothelial, at various gestational ages and developmental stages. The availability of scRNA-seq and the powerful inference trajectory tools enabled us to overcome these hurdles and investigate differentiation and segmentation processes at high resolution. It is plausible that secreted factors and cell–cell interactions from the various niches form the crosstalk allowing correct differentiation, such as in the case of stromal cells during nephrogenesis (Hum et al, 2014).

In summary, even though the developing human kidney is becoming increasingly difficult to obtain, it still holds a vast amount of knowledge that has yet to be unraveled and harnessed for authentication of hPSC-KO derivation protocols (Weissman et al, 2019). The derivation of hFKOs is straightforward and reproducible, strongly mimicking the developing hFK and holding great promise for advancing our understanding of development and disease.

# Methods

### Reagents and tools table

| Reagent/resource | Reference or source | Identifier or catalog number |
|---|---|---|
| **Experimental models** | | |
| Human fetal kidney tissue | Shamir Medical Center | IRB approval 0132-18-ASF |
| Human Adult Kidney | Sheba Medical Center | IRB approval SCM-5574-18 |
| Human iPSC-derived kidney organoids | Freedman et al, 2015 | WTC-11 iPSC line |
| Human ESC-derived kidney organoids | Takasato et al, 2016 | B123 hESC line |
| Wilms tumor xenograft | This study | |
| **Recombinant DNA** | | |
| – | – | – |
| **Antibodies** | | |
| Anti-NCAM1 (ERIC-1) | Santa Cruz | sc-106 |
| Anti-SIX2 | Proteintech | 11562-1-AP |
| Anti-PAX2 | BioLegend | 901001 |
| Anti-WT1 | Dako/Abcam | M3561/ab89901 |
| Anti-CDH6 | R&D Systems | MAB2715 |
| Anti-JAG1 | R&D Systems | AF599 |
| Anti-LHX1 | DSHB | 4F2 |
| Anti-MUC1 (EMA E29) | Cell Marque | 760-2762 |
| Anti-CDH1 | Cell Signaling | 2.40E + 11 |
| Anti-GATA3 | R&D Systems | AF2605 |
| Anti-HNF1B | Sigma-Aldrich | HPA002083 |
| Biotinylated LTL | Vector Laboratories | B-1325 |
| Anti-EPCAM | Merck Millipore/Abcam | CBL251/ab71916 |
| Anti-RET | R&D Systems | AF1485 |
| Anti-KI-67 | Monosan | MONX10283 |
| Anti-PODXL | Abcam | ab150358 |
| Anti-Synaptopodin | Abcam | ab224491 |
| Alexa Fluor® 488 donkey anti-rabbit IgG | Invitrogen | A21206 |
| Alexa Fluor® 555 donkey anti-rabbit IgG | Invitrogen | A31572 |
| Alexa Fluor® 647 Donkey Anti-Goat IgG | Abcam | ab150135 |
| Streptavidin, Alexa Fluor® 647 conjugate | Invitrogen | S21374 |
| **Oligonucleotides and other sequence-based reagents** | | |
| TaqMan® gene expression assays: CDH1 EPCAM SIX2 LHX1 JAG1 MUC1 AQP1 HEY1 HES1 GAPDH | Thermo Fisher Scientific | CHD1: Hs01023894_m1 EPCAM: Hs00158980_m1 SIX2: Hs00232731_m1 LHX1: Hs00232144_m1 JAG1: Hs01070032_m1 MUC1: Hs00410317_m1 AQP1: Hs00166067_m1 HEY1: Hs01114113_m1 HES1: Hs00172878_m1 GAPDH: Hs99999905_m1 |

| Reagent/resource | Reference or source | Identifier or catalog number |
|---|---|---|
| TaqMan® Fast Advanced Master Mix | Thermo Fisher Scientific | 4444557 |
| High-Capacity cDNA Reverse Transcription Kit | Thermo Fisher Scientific | 4374966 |
| Direct-zol™ RNA MiniPrep w/Zymo-Spin™ IIC Columns (Capped) | Zymogen Research | R2050 |
| TRIzol™ Reagent | Thermo Fisher Scientific | 15596026 |
| **Chemicals, enzymes, and other reagents** | | |
| BMP-7 | PeproTech Asia | |
| CHIR99021 | PeproTech Asia | SM-2520691-B |
| LDN-193189 | PeproTech Asia | 320-58-3 |
| A83-01 | PeproTech Asia | SML0788 |
| BMP-7 | PeproTech Asia | 120-03 |
| FGF-2 | PeproTech Asia | 100-18B |
| LIF | PeproTech Asia | AF-300-05 |
| Heparin, 25,000 I.E/5 mL | Roteximedica | – |
| ROCK inhibitor Y-27632 | PeproTech Asia | 129830-38-2 |
| EGF | PeproTech Asia | AF-100-15 |
| RSPO-1 | Cultrex / R&D Systems | R&D 4645-RS-025 |
| DAPT | PeproTech Asia | 2088055 |
| Dispase | Stem Cell Technologies | 7923 |
| Tryple Express | Gibco | 12605010 |
| BME (Cultrex) | R&D Systems | 3432-010-01 |
| Cryopreservation medium (NutriFreez D10) | Biological Industries | 05-713-1B |
| DMEM/F12 | Biological Industries | 01-170-1A |
| B-27 minus vitamin A | Gibco | 12587010 |
| 0.2% CD Lipid concentrate | Gibco | 11905031 |
| L-glutamine | Biological Industries | 03-020-1A |
| MEM Non-Essential Amino Acids Solution (100X) | Gibco | 11140035 |
| Sodium Pyruvate Mem 100 mM (Ce) | Gibco | 11360039 |
| Human FGF-10 | PeproTech Asia | 100-26-25 |
| Human FGF-basic | PeproTech Asia | 100-18B-1000 |
| Animal Free Human EGF | PeproTech Asia | AF-100-15-1000 |
| Collagenase, Type 4 | Worthington Biochemical | LS004189 |
| RBC Lysis Buffer (10X) | BioLegend | 420301 |
| Penicillin–Streptomycin Solution | Biological Industries | 03-031-1B |
| ITS liquid media supplement (100X) | Sigma-Aldrich | I3146 |
| CAS-Block™ | Invitrogen | 00-8120 |
| **Software** | | |
| FlowJo | BD | v7.6.5 |

| Reagent/resource | Reference or source | Identifier or catalog number |
|---|---|---|
| ZEN Blue | Zeiss | v3.7 |
| Fiji | ImageJ | v1.54 |
| OlyVia | Olympus | v4.1 |
| GraphPad Prism 9 | GraphPad Software | v9.0.0 |
| **Other** | | |
| NextSeq 500/550 High Output Kit v2.5 (75 Cycles) | Illumina | 20024906 |

## Ethics statement

This study was conducted according to the principles expressed in the Declaration of Helsinki. The study (under study numbers 0132-18-ASF, Human Fetal Kidneys, and SCM-5574-18, Human Adult Kidneys) was approved by the Institutional Review Boards of Shamir Medical Center and Sheba Medical Center. All patients provided written informed consent for sample collection and subsequent analysis.

## Generation of human fetal kidney organoids (hFKOs) from human fetal kidney tissue

### Tissue processing and single-cell suspension

1. **Obtain tissue:** Collect human fetal kidney (hFK) samples from elective abortions with proper ethical consent.
2. **Wash tissue**: Rinse the samples thoroughly with phosphate-buffered saline (PBS).
3. **Weigh and mince:** Weigh the tissue and mince into ~1-mm pieces using sterile scalpels.
4. **Enzymatic digestion:**
   - 500–1000 mg of minced tissue should be incubated in 20 mL of 0.1% collagenase IV (Worthington Biochemical) prepared in Iscove's Modified Dulbecco's Medium (IMDM; Biological Industries), corresponding to a tissue-to-solution ratio of 25–50 mg/mL.
   - Digest for 1.5–2 h at 37 °C with continuous shaking.
5. **Filter cell suspension**
   - Pass the digested tissue through a 100-μm cell strainer to obtain a single-cell suspension.
   - Wash the flask with PBS and pass through the cell strainer to collect leftover cells.
6. **Centrifugation and RBC lysis**
   - Centrifuge the cell suspension, 400 G for 5 min.
   - Resuspend the cell pellet in 15 mL of 1× red blood cell lysis buffer (BioLegend) for 12 min at room temperature, protected from light. Shake gently every 5 min. Solution should acquire a reddish-yellow hue due to RBC lysis.
   - Add 30 mL of PBS and centrifuge, 400 G for 5 min.
7. **Wash and count cells**
   - Wash cells with DMEM/F12 (Biological Industries).
   - Count viable cells.

**Tip:** Dissociated human fetal kidney cells can be frozen and used for generation of hFKOs at a later experiment.

### Organoid seeding and culture

8. **Prepare BME domes**
   - BME should be thawed 1 h before use and kept in ice in its liquid form.
   - Resuspend cells, carefully without introducing bubbles, at 1000 cells/μL in Cultrex BME (R&D Systems).
   - Plate 50-μL domes in 24-well culture plates.
9. **Solidify matrix:** Incubate at 37 °C for a short period until the BME solidifies (~10–15 min).

   **Tip:** To prevent spilling and deformation, plates can be incubated at 37 °C before adding BME, allowing the contact point of the dome to solidify.
10. **Add culture medium:** Add 250 μL of human nephron progenitor self-renewal (hNPSR) medium to each 24-well. Medium composition (modified from Li et al, 2016):

| Component name | Concentration | Supplier | Catalog number |
|---|---|---|---|
| DMEM/F12 | Base medium | Biological Industries | 01-170-1A |
| Penicillin–streptomycin | 1% (v/v) | Biological Industries | 03-031-1B |
| L-Glutamine | 1% (v/v) | Biological Industries | 03-020-1A |
| Non-essential amino acids (NEAA) | 1% (v/v) | Gibco | 11140035 |
| Sodium pyruvate | 1% (v/v) | Gibco | 11360039 |
| CD lipid concentrate | 0.2% (v/v) | Gibco | 11905031 |
| B-27 minus vitamin A | 1% (v/v) | Gibco | 12587010 |
| Insulin-transferrin-selenium (ITS) | 1% (v/v) | Gibco | I3146 |
| BMP-7 | 50 ng/mL | Peprotech Asia | 120-03P-50 |
| FGF-2 | 200 ng/mL | Peprotech Asia | 100-18B |
| Heparin | 1 μg/mL | Roteximedica | – |
| Y-27632 (ROCKi) | 10 μM | Peprotech Asia | 129830-38-2 |
| Human LIF | 10 ng/mL | Peprotech Asia | AF-300-05 |
| CHIR99021 | 1 μM | Peprotech Asia | SM-2520691-B |
| LDN-193189 | 50 nM | Peprotech Asia | 320-58-3 |
| A83-01 | 0.5 μM | Peprotech Asia | SML0788 |

**Tip:** ROCK inhibitor should be added to the culture medium at the initiation of culture, when single cells are present. Once organoids begin to form—typically 3 to 5 days after culture initiation—the ROCK inhibitor should be discontinued. The same principle applies during passaging—add ROCK inhibitor when organoids are disrupted and discontinue its use once organoid structures re-form.

### Passaging and maintenance

Every 2–3 weeks, depending on proliferation:

- Aspirate medium from wells containing BME domes with hFKOs.
- Wash wells with PBS and add 500-μL of ice-cold DMEM/F12 to each well.
- Mechanically disrupt hFKOs using a BSA-treated 1000-μL pipette tip. Avoiding BSA treatment results in significant organoid loss due to adherence.

  **Tip:** To increase shear stress, a 1000-μL pipette tip can be connected with a 100-μL tip fitted on top, allowing for more effective mechanical disruption.
- Aspirate the DMEM/F12 containing disrupted organoids and transfer it to a 15 mL tube containing 9 mL of ice-cold DMEM/F12. Collect organoids from up to six wells per tube, bringing the final volume to 12 mL.
- Incubate on ice for 40 min.
- Centrifuge in pre-chilled centrifuge, at 4 °C, 400 G for 5 min.
- Aspirate the supernatant, if a cloudy precipitate is present above the organoid pellet, gently aspirate it to remove any leftover BME.
- Resuspend in BME, passage at a 1:2 ratio.
- Plate 50-μL domes in 24-well culture plates and add hNPSR +ROCKi after the domes solidify.

### Cryopreservation

- Disrupt hFKOs in a similar fashion to passaging and resuspend in NutriFreez D10 (Biological Industries).
- Freeze using a cryo-cooler at –80 °C. Store at liquid nitrogen for long-term preservation.
- When thawing, resuspend hFKOs in BME at a 1:2 ratio.

## Generation of human embryonic stem cell-derived kidney organoids (hESC-KOs)

Organoids were differentiated according to the protocol by Takasato et al (Takasato et al, 2016) with minor modifications. In brief, cells at a density of $15 \times 10^3$ cells/cm$^2$ were grown on a Geltrex (Thermo Scientific) coated 25-cm$^2$ tissue culture flask with NutriStem hPSC XF Medium (Growth Factor-Free) (Biological Industries) supplemented with 8 μM CHIR99021 (TOCRIS) for 4 days. Medium was refreshed every 2 days. On day 5, medium was changed to NutriStem supplemented with 200 ng/mL FGF9 and 1 μg/mL Heparin (Stem Cell Technologies). Medium was refreshed every 2 days until day 7. On day 7, cells were lifted using Trypsin EDTA (0.05%, Biological Industries) and $3–5 \times 10^5$ cells were aliquoted per organoid and centrifuged at $400 \times g$ for 2 min to form a pellet. Pellets were then collected using a 200 μl wide-bore pipette tip and transferred intact onto a six-well transwell cell culture plate containing NutriStem with 5 μM CHIR99021. Following a 1-h incubation period, the medium was exchanged to NutriStem supplemented with 200 ng/ml FGF9 and 1 μg/ml Heparin. Medium containing growth factors was refreshed every 2 days for 5 days, then the factors were withdrawn, and organoids were grown for 2–4 additional days refreshing medium every 2 days. Whole-mount staining of hESC-KOs was done in a similar fashion to hFKOs.

## Immunofluorescence staining (IF)

### Whole-mount staining

hFKOs were cultured in imaging chambers (μ-Slide eight well, ibidi) and fixed for 15 min in 4% paraformaldehyde (PFA) solution. The PFA

solution was then replaced with 0.3% Triton X-100 in PBS for 30 min at room temperature for permeabilization. Samples were then washed three times with PBST (0.1% Triton X-100 in PBS) for 5 min. Blocking solution (3% BSA in PBST) was added for 1 h at room temperature, and this was followed by an overnight (ON) incubation at 4 °C in blocking solution supplemented with the following primary antibodies: anti-NCAM1 (1:50, ERIC-1, Santa Cruz), anti-SIX2 (1:50, 11562-1-AP, Proteintech), anti-PAX2 (1:250, 901001, BioLegend), anti-WT1 (1:00, 6F-H2, Dako, 1:100, ab89901, Abcam), anti-CDH6 (1:100, MAB2715, R&D Systems), anti-JAG1 (1:100, AF599, R&D Systems), anti-LHX1 (1:1, 4F2, DSHB), anti-MUC1 (1:500, EMA E29, Cell Marque), anti-CDH1 (1:200, 24E10, Cell Signaling), anti-GATA3 (1:300, AF2605, R&D Systems), anti-HNF1B (1:1000, HPA002083, Sigma-Aldrich), biotinylated *Lotus tetragonolobus* Lectin (LTL, 1:500, B-1325, Vector Laboratories), anti-EPCAM (1:500, CBL251, Merck Millipore, 1:500, ab71916, Abcam), anti-RET (1:100, AF1485, R&D Systems), anti-KI-67 (1:100, MONX10283, Monosan), and anti-PODXL (1:100, ab150358, Abcam). After three washes with PBST, secondary antibodies—Alexa488-conjugated anti-rabbit (1:500, A21202, Invitrogen), Alexa555-conjugated anti-mouse (1:500, A31570, Invitrogen), Alexa647-conjugated anti-goat (1:500, ab150135, Abcam), and Streptavidin-conjugated Alexa647 (1:500, S21374, Thermo Fisher)—and actin stain (1:1000, ab176759, Phalloidin-iFluor 647 Reagent, Abcam) were added in blocking solution and the samples incubated overnight at 4 °C. The samples were then washed in PBST, and DAPI-PBST solution was added (1:250, #62248, Thermo Fisher) for 10 min. The samples were washed again and imaged with a confocal microscope ZEISS LSM700; photo processing was done using ZEN blue. Widefield *z*-stack images were taken with a Nikon CSU-W1 Yokogawa Spinning Disk Field Scanning Confocal System and processed using Fiji (Schindelin et al, 2012).

### hFK tissue staining

In all, 5-μM paraffin sections were pre-treated using OmniPrep solution (pH 9.0) at 95 °C for 1 h in accordance with the manufacturer's protocol (Zytomed Systems). Blocking was done using Cas-Block solution (Invitrogen immunodetection, 00-8120) for 20 min and was followed by incubation with one of the primary antibodies listed above and detection using Alexa488-conjugated anti-rabbit, Alexa555-conjugated anti-mouse, Alexa647-conjugated anti-goat (1:1200), and Streptavidin-conjugated Alexa647 (1:1200) secondary antibodies for 60 min. Mounting medium containing DAPI (Dapi Fluoromount-G; SouthernBiotech, 0100-20) was applied. Slides were imaged using an Olympus IX83 fluorescence microscope and Olympus DP80 camera. Image processing was done using Olympus OlyVia software.

## Quantitative reverse transcription-PCR (qRT-PCR)

Total RNA was prepared using a Direct-zol RNA MiniPrep kit (Zymo Research) according to the manufacturer's instructions. Quantitative real-time PCR was performed as previously described (Dalerba et al, 2011). TaqMan gene expression primers and probes were purchased from Thermo Fisher Scientific.

## Flow cytometry

Fresh hFKs were dissociated into single cells, and $0.5 \times 10^5$ cells were suspended in FACS buffer (0.5% BSA in PBS). The cells were then incubated with a primary antibody, anti-CD56 (NCAM1)-APC (eBioscience) or anti-CD326 (EPCAM)-PE (eBioscience), or an isotype control. Cell viability was tested using 7-aminoactinomycin D viability staining solution (eBioscience). Cell labeling was detected using a FACSCalibur flow cytometer (BD PharMingen). Flow cytometry results were analyzed using FlowJo analysis software (version 7.6.5).

## Magnetic cell sorting (MACS)

hFKOs were grown for two passages (P2, 8 weeks of culture) and dissociated into single cells, which were isolated by positive selection of NCAM1 (CD56 Microbeads, 130-097-042, Miltenyi Biotec) according to the manufacturer's instructions. Positive and negative fractions were collected and cultured in BME supplemented with hNPSR culture medium. Enrichment of NCAM1+ cells was validated by flow cytometry (Fig. EV4D).

## Bulk RNA sequencing

### Generation of human fetal kidney organoid (hFKO) RNA samples

Culture medium was aspirated, and 1 mL of warm 1 mg/mL Dispase (# 07923, Stem Cell Technologies) was added to each well of a 24- well plate containing a BME droplet with hFKOs. The droplets were then gently broken down by pipetting up and down using a 1000-μl pipette tip and transferred to a new well to avoid contamination with cells growing outside of the BME droplet. After a 40-min incubation at 37 °C, the BME was dissolved, and the hFKOs were transferred to 1.5 mL Eppendorf tubes and washed with DMEM F12. The medium was then aspirated and replaced with 300 μL of Trizol (Thermo Fisher Scientific, 15596018) per tube and stored at –80 °C. Total RNA was prepared using a Direct-zol RNA MiniPrep kit (Zymo Research) according to the manufacturer's instructions.

### Generation of adult kidney organoid (AKO) RNA samples

Tissue processing was done similarly to that for hFKs. After embedding of AK cells in BME, a previously described kidney tuboloid medium (Schutgens et al, 2019), AKO medium, was added, composed of DMEM F12 supplemented with 1% penicillin–streptomycin 100 M, 1% L-glutamine (both purchased from Biological Industries), 1% non-essential amino acids (Invitrogen), 10% R-spondin1-conditioned medium (Cultrex, HA-R-Spondin1-Fc 293 T Cells), 1% B-27 supplement (Gibco), 1 mM *N*-acetyl cysteine (NAC, Sigma), 100 ng/mL FGF-10, 50 ng/mL EGF, 1 μg/mL heparin, 10 μM Y-27632, and 5 μM A83-01 (the latter all purchased from Peprotech Asia). AKOs were mechanically disrupted with a BSA-treated 1000-μl pipette tip and passaged at a 1:2 ratio every 2–3 weeks, depending on the proliferation rate. RNA was prepared in the same method as the hFKOs.

### Generation of day 10 and day 18 iPSC-KO RNA samples

A protocol similar to that described in a previous publication (Freedman et al, 2015) was used. In brief: on day 0, iPSCs were dissociated to single cells with Accutase (Stem Cell Technologies), resuspended in mTeSR1 (85850, Stem Cell Technologies) supplemented with 10 mM of the Rho kinase inhibitor Y-27632 (Tocris Bioscience), and seeded in a 24-well plate pre-coated with Geltrex (A1413201, Thermo Fisher) at a low density (1000–2000 cells per well). The next day, the medium was replaced with 500 mL mTeSR1 + 1.5% Geltrex. On day 2, the medium

was replaced with 500 mL fresh mTeSR1. On the evening of day 3, cells were pulsed with 12 mM CHIR99021 (Tocris Bioscience) in Advanced RPMI + Glutamax (Life Technologies) for 36 h and then switched to organoid maintenance medium (RB), consisting of Advanced RPMI with 1× Glutamax and 1× B-27 Supplement (Life Technologies). RB was changed 3 days later and every 2–3 days thereafter. For sampling, at room temperature, medium was aspirated from the cells and replaced with 500 μL of Trizol (Thermo Fisher Scientific, 15596018) per well of a 24-well plate. Cells were homogenized by scraping the well with the pipette tip and pipetting up and down several times and then frozen until further processing.

### Sequencing

Sequencing libraries were prepared starting from the frozen RNA samples using MARS-seq (Illumina). Single reads were sequenced on 1 lane of an Illumina NextSeq sequencer.

### Bioinformatics

Poly(A/T) stretches and Illumina adapters were trimmed from the reads using cutadapt (Kechin et al, 2017); resulting reads shorter than 30 bp were discarded. Reads were mapped to the *Homo sapiens* reference genome GRCh38 using STAR (Dobin et al, 2013), supplied with gene annotations downloaded from Ensembl (and with EndToEnd option and outFilterMismatchNoverLmax was set to 0.04). Expression levels for each gene were quantified using htseq-count (Anders et al, 2015).

### RNA-seq data analysis

Data normalization and differential gene expression were done by DESeq2 (Love et al, 2014). The R packages pheatmap and ggplot2 were used for data visualization. The Gene Set Enrichment Analysis tool was used to find gene sets that were enriched in two gene lists ranked by their $\log_2$ (fold change) ratio (Subramanian et al, 2005). Ontology analysis was performed using ClusterProfiler (Yu et al, 2012) and visualized with GOplot (Walter et al, 2015). Gene enrichment analysis was performed using ToppGene (Chen et al, 2009), and revealed enrichment for annotations related to kidney development, proliferation, structure, and function.

## Notch inhibition in hFKOs

hFKOs were generated as described above. After 3 days of initial culture, DAPT, reconstituted in DMSO, was added to the culture medium at 5 μM final concentration; this medium was refreshed every 2 days. hFKOs were grown for 2–3 weeks under Notch inhibition next to control wells in which the culture medium was supplemented with DMSO only. Optimal DAPT concentration was chosen based on downregulation of Notch downstream target HES1 while retaining viable hFKOs (Fig. EV6F).

## Single-cell RNA sequencing

### Preparation of single-cell suspension

Culture medium was aspirated, and 1 mL of warm 1 mg/mL Dispase (# 07923, Stem Cell Technologies) was added to each well of a 24-well plate containing a BME droplet with hFKOs. The droplets were then gently broken down by pipetting up and down using a treated 1000-μl pipette tip (obtained by pipetting 1% BSA in PBS up and down five times in standard pipette tips) to prevent hFKOs fragments from

sticking to the pipette tip. After a 40-min incubation at 37 °C, the BME was dissolved and the hFKOs were transferred to 15-mL tubes containing ice-cold DMEM F12, centrifuged, and resuspended in 3 mL 1× TrypLE express (#12605010, Gibco) supplemented with 10 μM ROCKi. The tubes were incubated at 37 °C for a 10-min cycles, after which the hFKOs were disturbed by pipetting up and down 15 times and 10 μL of cell suspension was checked under a microscope to determine whether a single-cell suspension had been achieved. This process was then repeated until that was accomplished; two or three cycles are usually enough to break down the hFKOs to single cells without affecting cell viability.

### Single-cell RNA-seq library preparation and sequencing

hFKO cells were harvested, washed, and resuspended in PBS supplemented with 0.5% BSA to achieve an optimal concentration of approximately 1000 cells/μL for loading onto the Next GEM Chip (10x Genomics). Libraries were prepared using the 10x Genomics Chromium Controller in conjunction with the single-cell 3' v3.1 kit, protocol revision E. cDNA synthesis, barcoding, and library preparation were carried out according to the manufacturer's instructions. Briefly, cDNA amplification was performed for 11 cycles and sample index PCR was performed for 12 cycles using Chromium i7 Sample Indices. The resulting libraries were quantified and analyzed with Qubit and TapeStation and were sequenced on the NextSeq 500 platform (Illumina), following the manufacturer's protocol, using a NextSeq 500/550 High Output Kit v2.5 (75 cycles) kit (Illumina).

### Pre-processing

Cell ranger (v3) pipeline was used to perform sample demultiplexing to produce FASTQ files for each sample. Samples were aligned to the GRCh38 reference human genome, and cell barcodes were filtered so that only barcodes connected to unique molecular identifiers and gene IDs were used in a matrix file compatible with Seurat.

### Quality control

The standard QC workflow as presented in Seurat vignettes from the Satija Lab were followed. In brief, low-quality cells, empty droplets with low gene counts, and doublets/multiplets with abnormally high gene counts were filtered out. In addition, cells with high counts of mitochondrial genes were excluded. The data were then normalized, highly variable features were identified, and scaling and linear dimension reduction were performed.

### Clustering analysis

We utilized Seurat's (Butler et al, 2018; Satija et al, 2015) (v4.3.0) FindNeighbors(), and FindClusters() to produce a UMAP containing the various single-cell clusters. The resolution of FindClusters() was determined with the Clustree package (Zappia and Oshlack, 2018), which generates cluster trees with branches depicting known markers of kidney development and mature nephron compartments. For the first characterization experiment, datasets of 2 hFKs, from weeks 15 and 20 of gestation, containing 13,370 cells, and resolution of 0.5 was chosen which produced 16 clusters. Differentially expressed genes were found for each cluster using FindAllMarkers(), with the parameters min.pct = 0.25, logfc.threshold = 0.25, and p_val_adj < 0.05. After defining the subset constituting the nephrogenic zone, we re-clustered 10,351 cells with 0.5 resolution. For the Notch inhibition experiment, hFKOs from the same single hFK were grown with and without DAPT and merged into a single dataset containing 11,869 Notch-inhibited hFKO cells and

7418 DMSO control cells, clustered with a resolution of 0.5. After sub-setting the nephrogenic zone, we re-clustered 6354 cells with a resolution of 0.3. The DT/CNT-DAPT cluster was isolated, and 603 cells were re-clustered at a resolution of 0.2.

### Pseudotime analysis

We utilized monocle3 (Trapnell et al, 2014; Qiu et al, 2017) to perform pseudotime analysis. We input the Seurat objects containing the nephrogenic cluster from each experiment, set the root as the earliest biological timepoint known (NPCs or RVs), and ran the pseudotime algorithm to generate cellular trajectories. To compare trajectories between multiple conditions in the Notch inhibition experiment, we utilized the Condiments(Roux de Bézieux et al, 2024) package, based on separate trajectory inference by the slingshot package (Street et al, 2018), to create an imbalance plot and differential topology plot. We utilized Monocle's find_gene_module() to cluster DE genes into modules that are co-expressed across clusters and conditions.

## Comparison to other datasets

Publicly available bulk RNA datasets of iPSC-derived KOs used for comparison to hFKOs: GSE164648, GSE70101. For the single-cell dataset for week 17 of gestation, cortical hFKs containing the metanephric zone and PDX-Wilms tumor datasets were compared to the dataset containing hFKOs from two hFKs by Seurat's MapQuery() function, which locates anchors between reference and test datasets. The DevKidCC package (Wilson et al, 2022) was used to compare our hFKO single-cell datasets to hFK references as well as PSC-KO datasets, according to the instructions published with the package.

## Statistical analysis

Data are presented as mean ± standard deviation (SD) unless otherwise indicated. A $P$ value < 0.05 was considered statistically significant. All statistical analyses were performed using GraphPad Prism 9 (GraphPad Software, San Diego, CA).

## Graphics

Synopsis and additional schemes were created with BioRender.com.

## Data availability

Bulk and single-cell RNA sequencing data generated in this study include two bulk RNA-seq experiments comparing P0–P2 human fetal kidney organoids (hFKOs) and d10/d18 iPSC-derived kidney organoids with P0–P5/P6 hFKOs. In addition, two single-cell RNA-seq experiments were performed: one characterizing two independent P0 hFKO samples, and another comparing hFKOs treated with the γ-secretase inhibitor DAPT to DMSO-treated controls. All datasets are deposited in the GEO repository under accession numbers GSE271315, GSE300013, GSE300166, GSE300309.

The source data of this paper are collected in the following database record: biostudies:S-SCDT-10_1038-S44318-025-00504-2.

## Peer review information

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

## Acknowledgements

We thank Rami Khosravi at the Gray Faculty of Medical & Health Sciences Research Infrastructure Core Facilities of Tel Aviv University, and Hila Kobo from The Rosalie and Harold Rae Brown Cancer Research Core Facility, Faculty of Life Sciences, Tel Aviv University, for their crucial assistance in performing the single-cell RNA sequencing experiments. We thank Nadav Bakst for his assistance with acquiring hESC-derived kidney organoids samples for immunostaining. MN was supported by a PhD scholarship from the Yoran Institute for Human Genome Research at Tel Aviv University. This work is supported by the United States–Israel Binational Science Foundation (BSF, grant No. 2017274), the Lisa and David Pulver Foundation, The Euro-Asian Jewish Congress, Israel Science Foundation (ISF) no. 2071/17 and no. 3155/21, the Israel Precision Medicine Partnership (IPMP) no. 1814/20, Israel Ministry of Science no. 3-16220, and Ezvonot no. 20210231.

## Author contributions

**Michael Namestnikov**: Conceptualization; Data curation; Formal analysis; Validation; Investigation; Visualization; Methodology; Writing—original draft; Writing—review and editing. **Osnat Cohen-Zontag**: Data curation; Software; Formal analysis; Validation; Investigation; Visualization; Project administration. **Dorit Omer**: Methodology; Project administration. **Yehudit Gnatek**: Investigation; Methodology. **Sanja Goldberg**: Methodology. **Thomas Vincent**: Resources; Visualization. **Swati Singh**: Investigation. **Yair Shiber**: Resources. **Tal Rafaeli Yehudai**: Resources. **Hadas Volkov**: Data curation; Software. **Dani Folkman Genet**: Resources. **Achia Urbach**: Resources. **Sylvie Polak-Charcon**: Investigation; Visualization. **Igor Grinberg**: Investigation; Visualization. **Naomi Pode-Shakked**: Resources; Writing—review and editing. **Boaz Weisz**: Resources. **Zvi Vaknin**: Resources. **Benjamin S Freedman**: Resources; Writing—review and editing. **Benjamin Dekel**: Conceptualization; Supervision; Funding acquisition; Methodology; Writing—original draft; Project administration; Writing—review and editing.

Source data underlying figure panels in this paper may have individual authorship assigned. Where available, figure panel/source data authorship is listed in the following database record: biostudies:S-SCDT-10_1038-S44318-025-00504-2.

## Disclosure and competing interests statement

The authors declare no competing interests.

# Expanded View Figures

**Figure EV1. human fetal kidney organoid (hFKO) phenotype under various culture medium.**

(A) Left: schematic representation of the different phenotypes derived from each medium. hNPSR medium forms hFKOs that are complex and convoluted, while AKOM medium drives a cystic phenotype. Middle panel: hFKOs under hNPSR form complex, convoluted structures (dashed white lines). Right: hFKOs under AKOM show a simple cystic phenotype, similar to that of adult kidney-derived tubuloids. Scale bars, 100 μm. (B) Representative widefield image of a BME droplet containing hFKOs with convoluted phenotype under hNPSR. Scale bars, 500 μm. (C) Representative widefield image of a BME droplet containing hFKOs showcasing a cystic phenotype under AKOM. Scale bars, 500 μm. (D) Electron microscopy of hFKOs. Left: Microvilli (black arrows) are present in the apical lumen of hFKOs, along with basement membrane (BM). Middle: Tight junctions (white arrows) and lumens with microvilli (dashed white arrow) between cells in hFKOs. Left: Primary cilia (black arrow), often present in proximal and distal tubules as well as collecting ducts in the mature kidney. (E) hFKOs contain KI-67$^+$ cells which allow their long-term proliferation. qRT-PCR of P0 and P6 hFKOs reveal increased *CDH1* expression and maintenance of *EPCAM* expression, suggesting a stable renal epithelial phenotype. $n = 3$; expression levels in P0 were used to normalize data. Data were calculated as average ± SD. *$P < 0.05$., immunostaining of P2 and P6 hFKOs confirm the maintenance of an epithelial identity. Scale bars, 20 μm (top right), 50 μm (bottom panels). (F) Not all hFKOs swell under forskolin treatment since the culture is enriched with early tubular epithelial cells, which putatively do not possess the correct channels for water absorption. Scale bars, 500 μm.

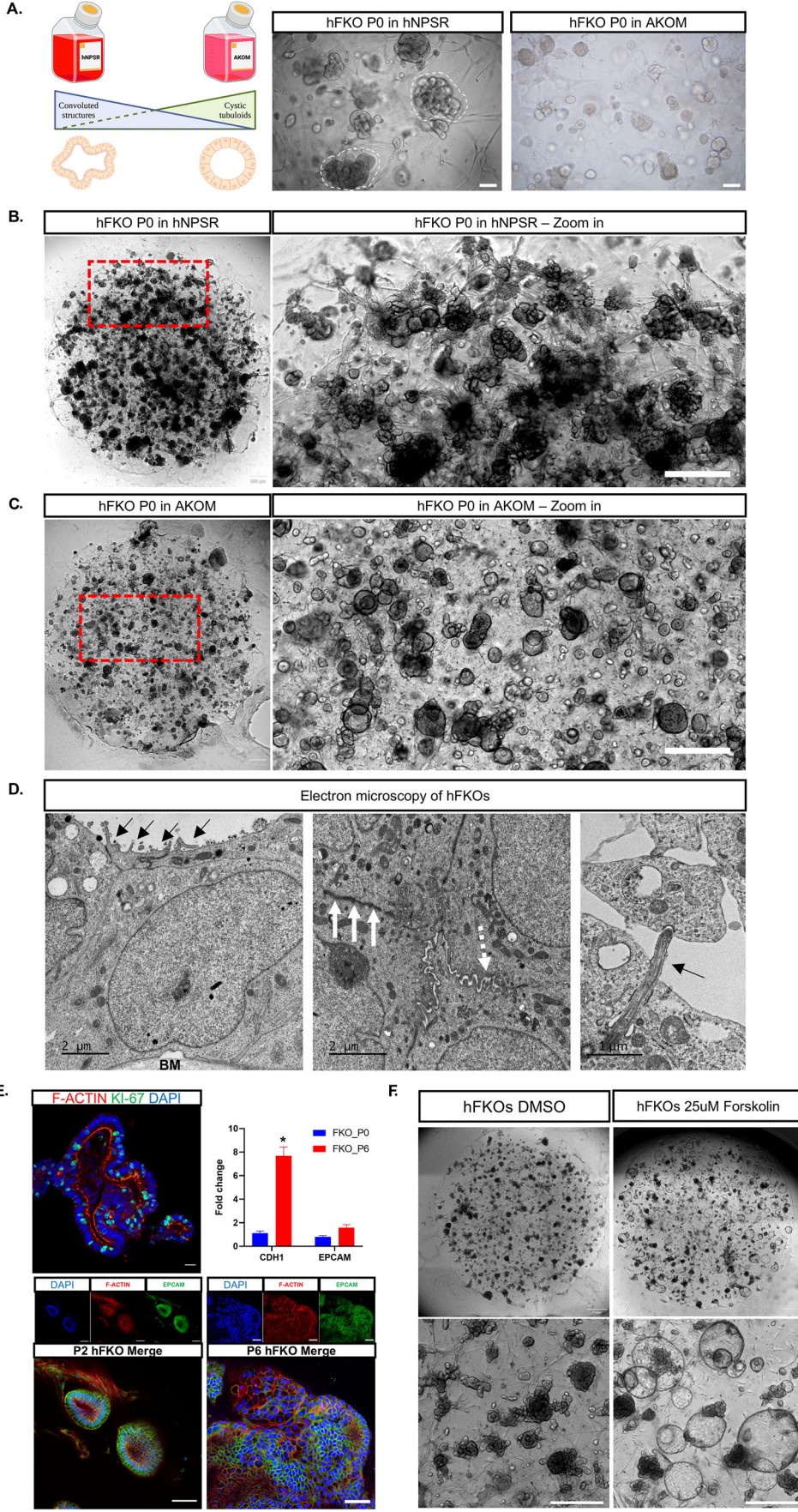

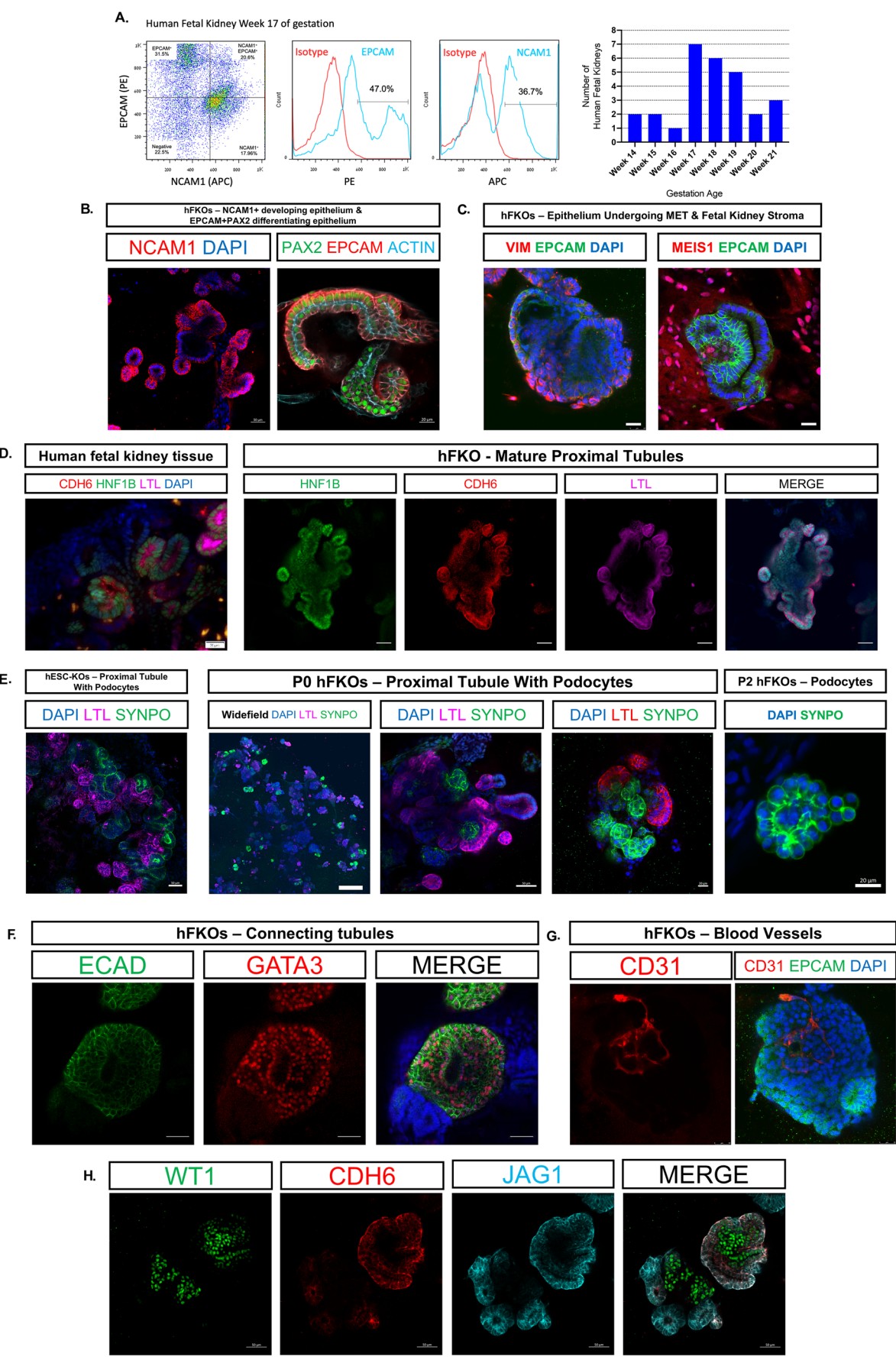

◀  **Figure EV2.  Developmental hierarchy is preserved in hFKOs.**

(**A**) Flow cytometry analysis of composition of freshly dissociated hFKs containing NCAM1$^+$EPCAM$^-$ early epithelial progenitors as well as NCAM1$^+$EPCAM$^+$ committed early differentiated epithelium. Distribution of gestational age of hFKs received for research; weeks 17–19 were the most common. (**B**) Morphological difference between NCAM1$^+$PAX2$^+$ early developing nephrons and EPCAM$^+$ mature differentiated epithelial. NCAM1$^+$ structures are aggregates, while EPCAM$^+$ structures are more convoluted and tubular in nature. (**C**) hFKOs with EPCAM$^+$VIM$^+$ cells undergoing mesenchymal-to-epithelial transition (MET) and also MEIS1$^+$ fetal kidney stromal cells surrounding the epithelial structures. (**D**) Mature proximal tubule structures in hFKOs expressing HNF1B, CDH6, and LTL. (**E**) Left, human embryonic stem cell-derived kidney organoids stained for LTL and synaptopodin (SYNPO). Middle, P0 hFKOs with podocytes expressing SYNPO accompanied by LTL$^+$ proximal tubules. Left, P2 hFKOs expressing SYNPO. (**F**) ECAD$^+$GATA3$^+$ connecting tubules in hFKOs. (**G**) CD31$^+$ endothelial cells form small blood vessels in EPCAM$^+$ hFKOs. (**h**) hFKOs consist of multiple lineages, including WT1$^+$CDH6$^+$ proximal tubules and JAG1$^+$ medial and distal lineages.

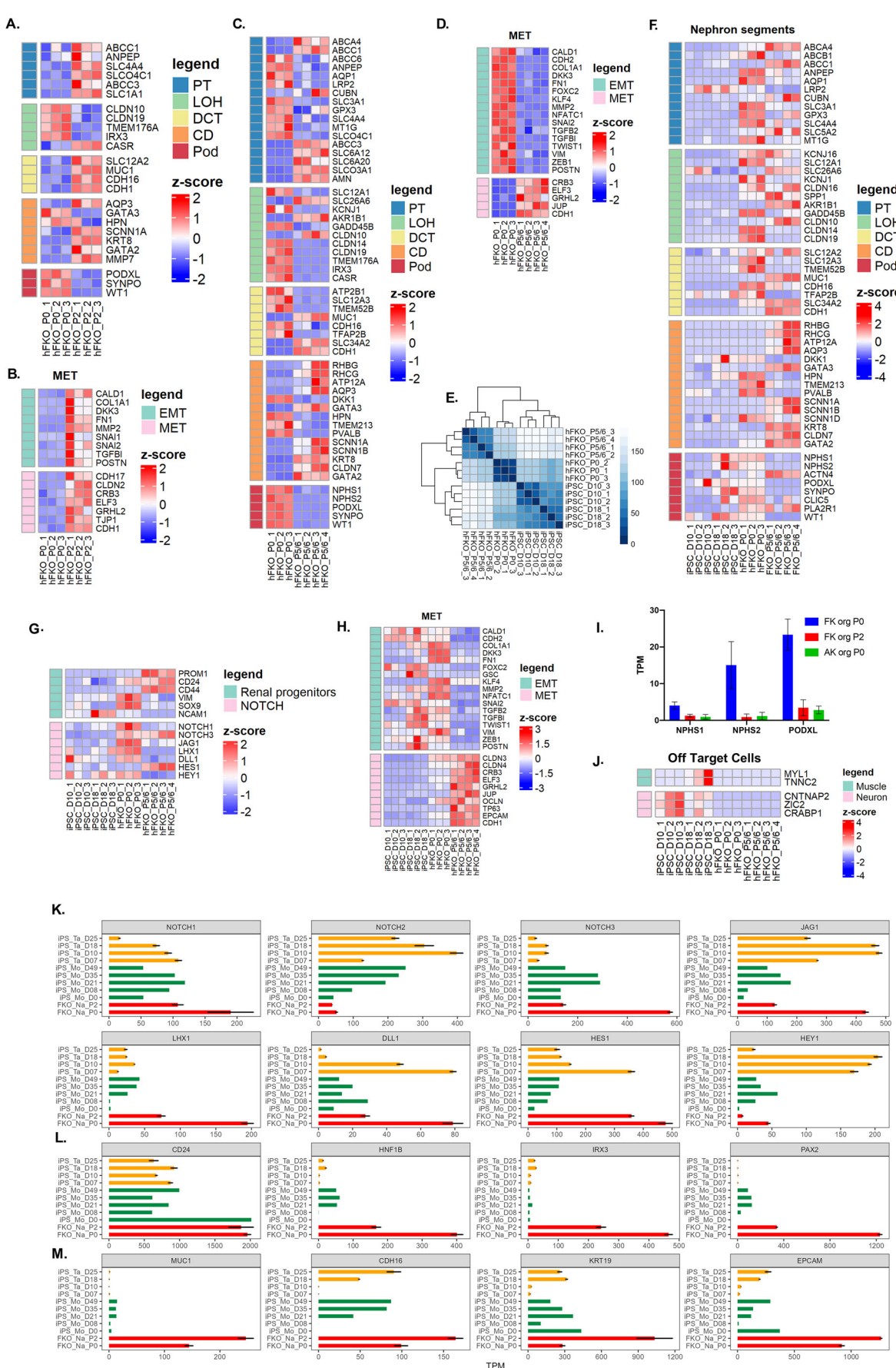

◄ **Figure EV3. Bulk RNA sequencing of hFKOs shows superior expression of kidney development genes.**

(A) Nephron segment markers increase in P2 of hFKOs, indicating maturation process in vitro. (B) EMT and MET markers indicate a dual process mesenchymal proliferation and epithelial transition, suiting the self-renewal of NPCs and differentiation of early nephrons. (C) Heatmap comparing the expression of segment-specific markers between early (P0) and late (P5/P6) hFKOs. (D). Comparison of expression of EMT and MET markers in P0 and P6 hFKOs. (E) Distance plot of human fetal kidney organoids (hFKO), and iPSC-derived kidney organoids (iPSC). (F) Nephron segment markers of P0 and P6 hFKOs vs D10 and D18 iPSC-derived kidney organoids (iPSC-KOs). (G) Renal progenitor and Notch signaling pathways related genes expression in P0 and P6 hFKOs vs D10 and D18 iPSC-KOs. (H) Comparison of expression of EMT and MET markers in P0 and P6 hFKOs vs D10 and D18 iPSC-KOs. (I) Podocyte marker expression in P0 hFKOs versus P0 Adult kidney organoids (AK org). hFKOs contain all nephron segments in comparison to AKOs which contain only the tubular segments. (J) Comparison of off-target marker expression between iPSC-KOs and hFKOs, expression of muscles markers, MYL1 and TNNC2 and neuron markers CNTNAP2, CRABP1 and ZIC2 are higher in iPSC-KOs. (K) Key Notch markers expression (in transcripts per million, TPM) in P0 and P2 hFKOs versus other prominent hPSC-KO differentiation protocols, Morizane et al (Mo) and Takasato et al (Ta), at various timepoints (Mo: day 8, 21,35 and 49. Ta: day 7, 10, 18 and 25). Notch1, Notch3, JAG1, LHX1, DLL1 and HES1 are expressed on par with hPSC-KOs or expressed higher in hFKOs. (L) Expression of key epithelial progenitor markers such as CD24, HFN1B, IRX3 and PAX2 in hFKOs in comparison to other hPSC-KO protocols, HNF1B, IRX3 and PAX2 are expressed higher in hFKOs, even in P2, indicating that the hFKO culture retains a population of epithelial progenitors even after 8 weeks of culture. In comparison to hFKOs, PAX2 and HNF1B are quickly downregulated in hPSC-KOs. (M) Expression of nephron epithelium markers, MUC1, CDH16, KRT19 and EPCAM, is increased in P2 hFKOs, indicating differentiation and maturation processes similar to the native kidney.

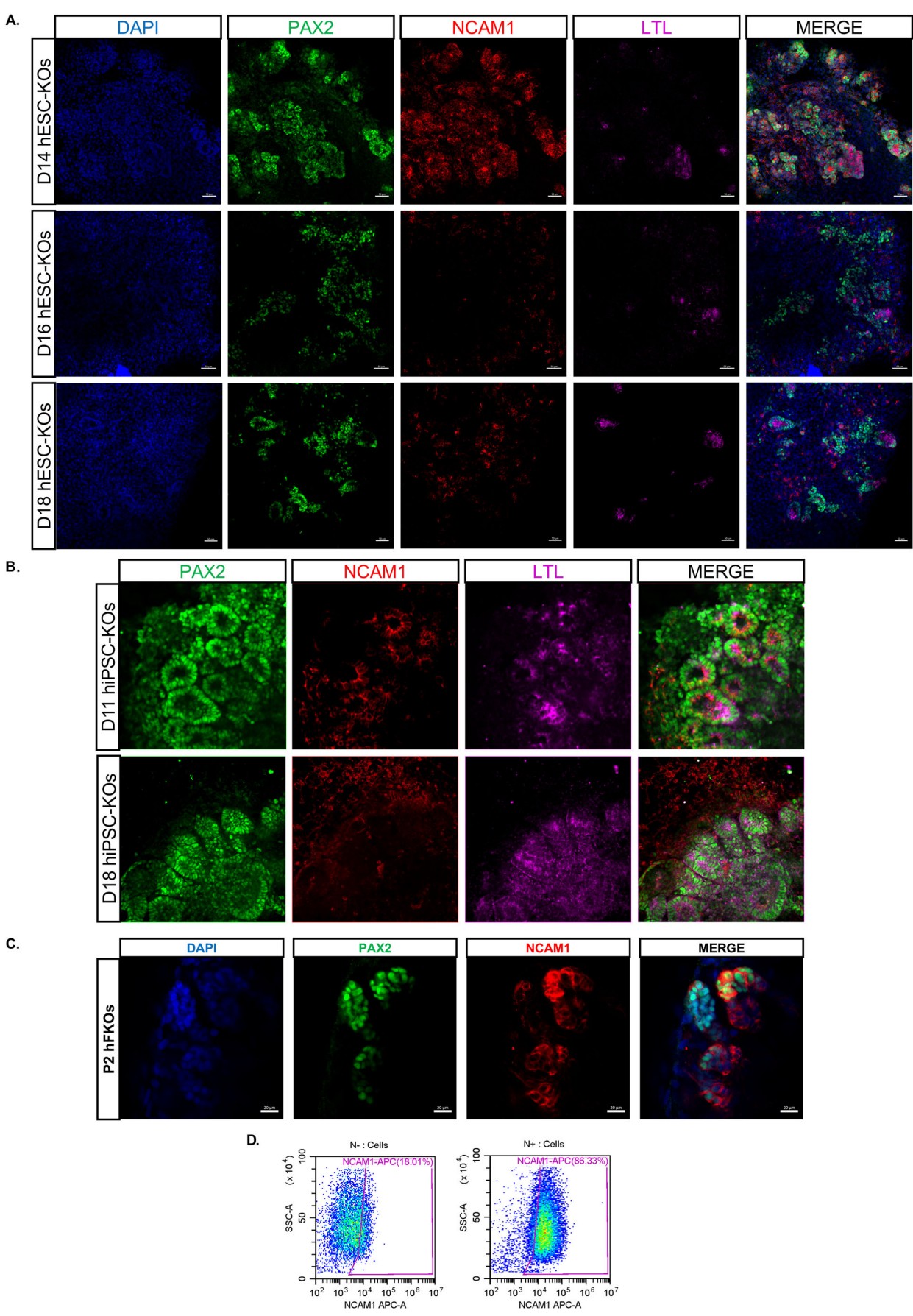

◀ **Figure EV4.  Nephron progenitor markers NCAM1 and PAX2 are less abundant as hPSC-derived kidney organoids mature.**

(A) Immunofluorescent imaging of human embryonic stem cell-derived kidney organoids (hESC-KOs), expressing many NCAM1+ and PAX2+ structures, containing nephron progenitor cells at Day 14 of differentiation. These structures become less abundant as the culture progresses, even after an additional two days (D16). As the differentiation progresses, PAX2+ cells become organized into tubular structures and decrease in number. scale bar 50 μm. (B) IF imaging of hiPSC-derived kidney organoids. NPC populations co-expressing NCAM1 + PAX2+ diminish as the organoid differentiates and NCAM1 cells become unorganized and are more abundant in the stroma of the organoid. (C) in P2 hFKOs, after 8 weeks of culture, NCAM1 + PAX2+ co-expression is evident in epithelial progenitors. Scale bar 20 μm. (D) Flow cytometry of positive and negative fractions of NCAM1+ after magnetic sorting.

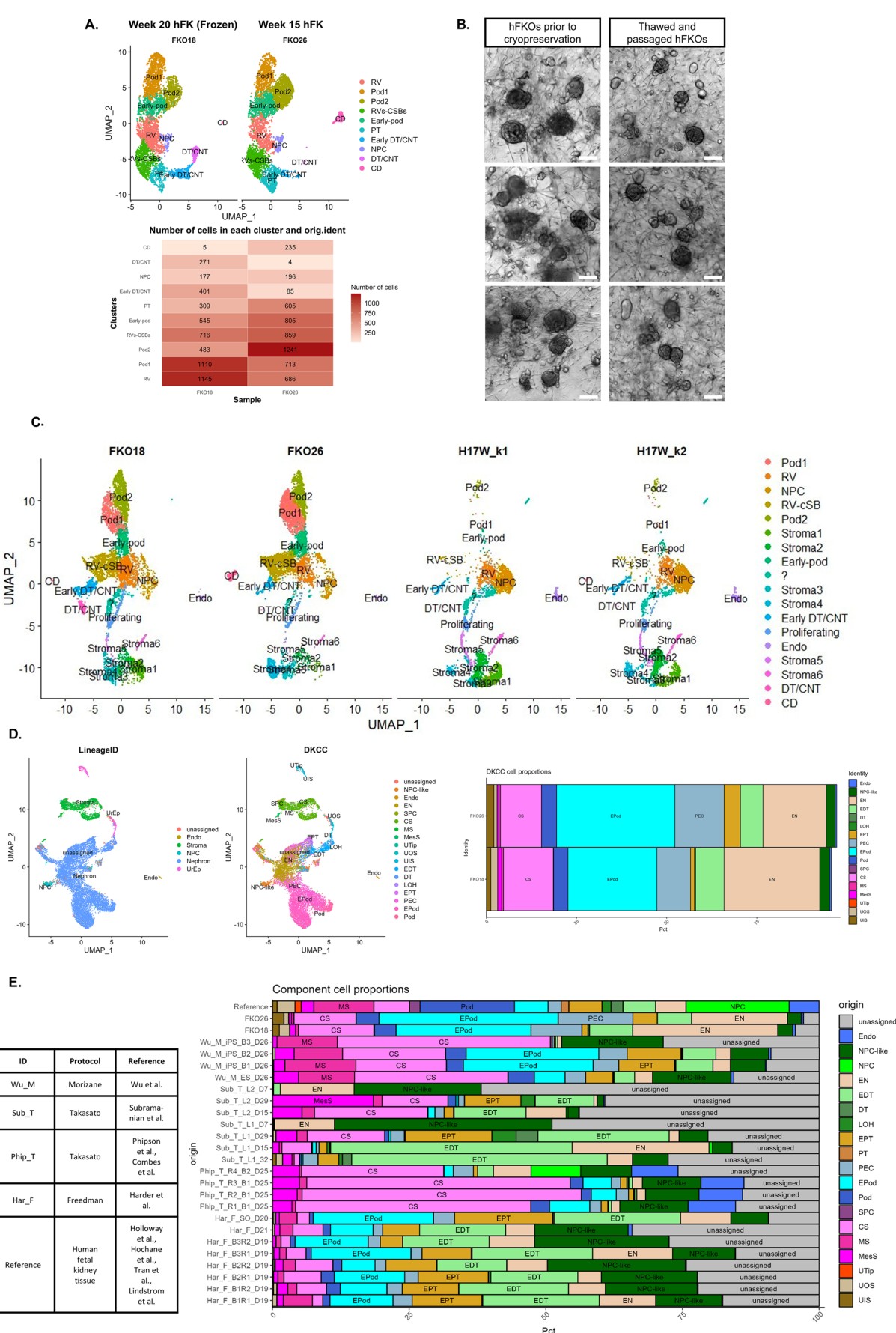

◀ **Figure EV5.  Single-cell RNA sequencing of hFKOs reveals developmental processes similar to those of native fetal kidney.**

(**A**) UMAP of each hFKO sample used in the scRNA-seq dataset after nephrogenic sub-setting. The two samples exhibit similarity in clusters, even though one hFKO sample originated from thawed hFK cells. Right: Table depicting number of cells in each cluster. (**B**) hFKO cells cryopreserved compared to hFKO cells after thawing and continued culture, no significant change of morphology was detected. (**C**) Comparison of hFKOs scRNA-seq dataset to human fetal kidney cortex dataset, matching NPCs and RVs. (**D**) Unbiased classification of hFKOs clusters with the DevKidCC tool. Clusters are classified similarly to initial classification made using markers depicted in the literature. (**E**) Comparison of nephrogenic clusters and their proportions between hFKOs, human fetal kidney tissue, and kidney organoids derived from PSCs using prominent organoid differentiation protocols. ID, protocols, and references used to produce the plot are depicted in the table at left. Origin annotations: (UTip), outer stalk (UOS), inner stalk (UIS), stromal progenitor cells (SPC), cortical stroma (CS), medullary stroma (MS), mesangial cells (MesS), endothelium (Endo), nephron progenitor cells (NPC), early nephron (EN), early distal tubule (EDT), distal tubule (DT), Loop of Henle (LOH), early proximal tubule (EPT), proximal tubule (PT), parietal epithelial cells (PEC), early podocytes (EPod) and podocytes (Pod).

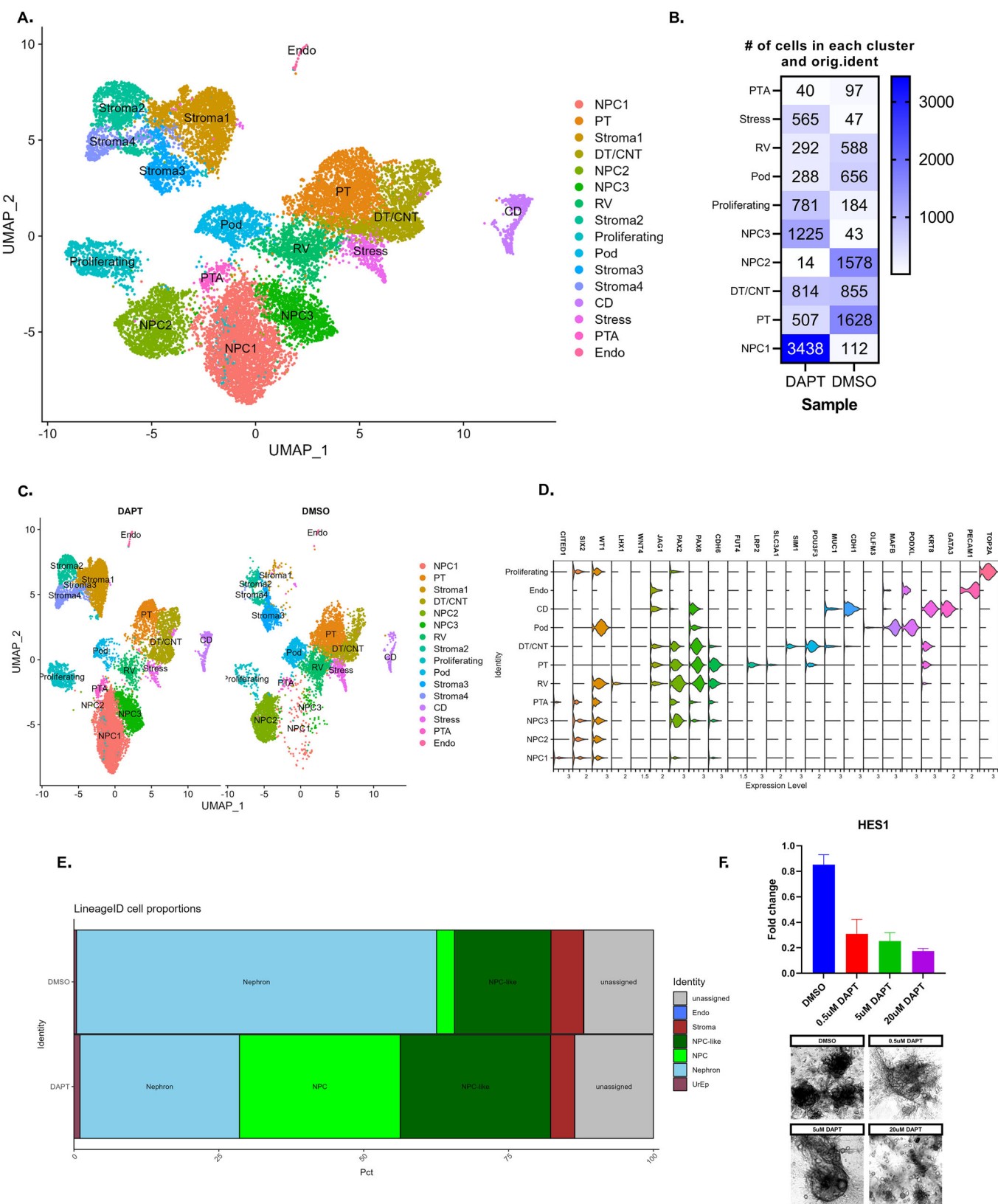

◀ **Figure EV6. Proximal tubule lineage is hampered under Notch inhibition in hFKOs.**

(A) UMAP of DAPT and control hFKOs clusters. NPC, nephron progenitors; RV, renal vesicles; Pod, podocytes; DT/CNT, distal/connecting tubule; PT, proximal tubule; PTA, pre-tubular aggregates; endo, endothelial cells, CD, collecting ducts. (B) Each treatment leads to different proportions of cells in each cluster, an increased NPC population under DAPT inhibition. and decreased PT and RV cells. (C) UMAP split by sample identity, DAPT and control. hFKOs under DAPT treatment have more NPCs, which are retained in a progenitor state under Notch inhibitions. (D) Violin plots of the expression of markers used in classifying clusters. (E) Unbiased classification of clusters in hFKOs under DAPT inhibition and control (DMSO) with the DevKidCC tool. (F) DAPT inhibits HES1 expression in a dose-dependent manner, as the concentration of DAPT increases, the fold change in HES1 expression decreases. Morphological changes, such as reduction in organoid complexity, begin to be evident at concentrations of 5 μM DAPT while no significant change in expression is observed in higher concentrations.

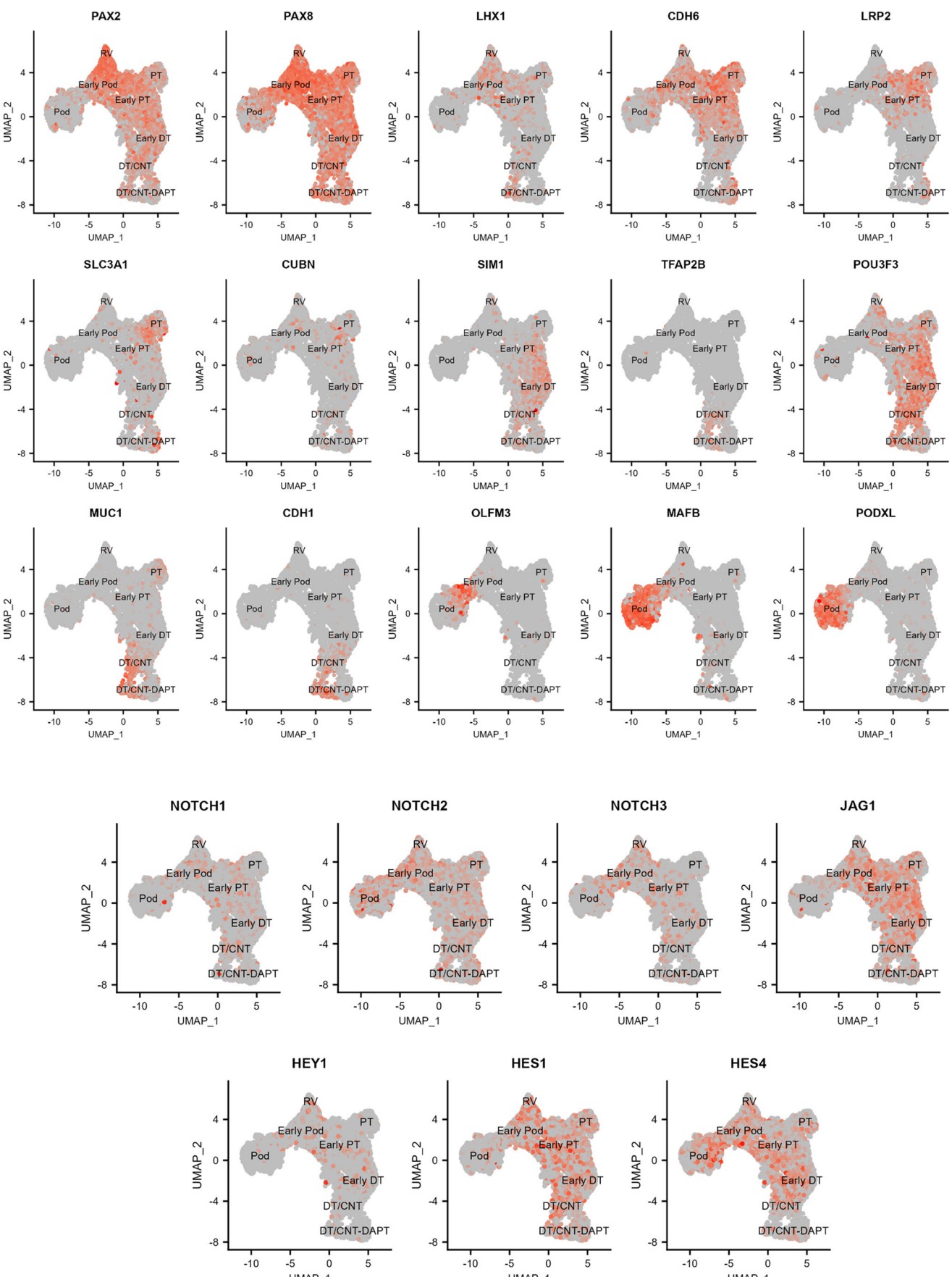

◀ **Figure EV7. Proximal tubule lineage is hampered under Notch inhibition in hFKOs.**

Feature plots of key lineage markers used to infer differentiation states: for example, *TFAP2B* and *POU3F3* are marker of early distal lineage and *CDH1* and *MUC1* are mature distal tubule markers, *CDH6* is a distal early proximal tubule marker, and *LRP2* and *SLC3A1* are mature proximal tubule markers.

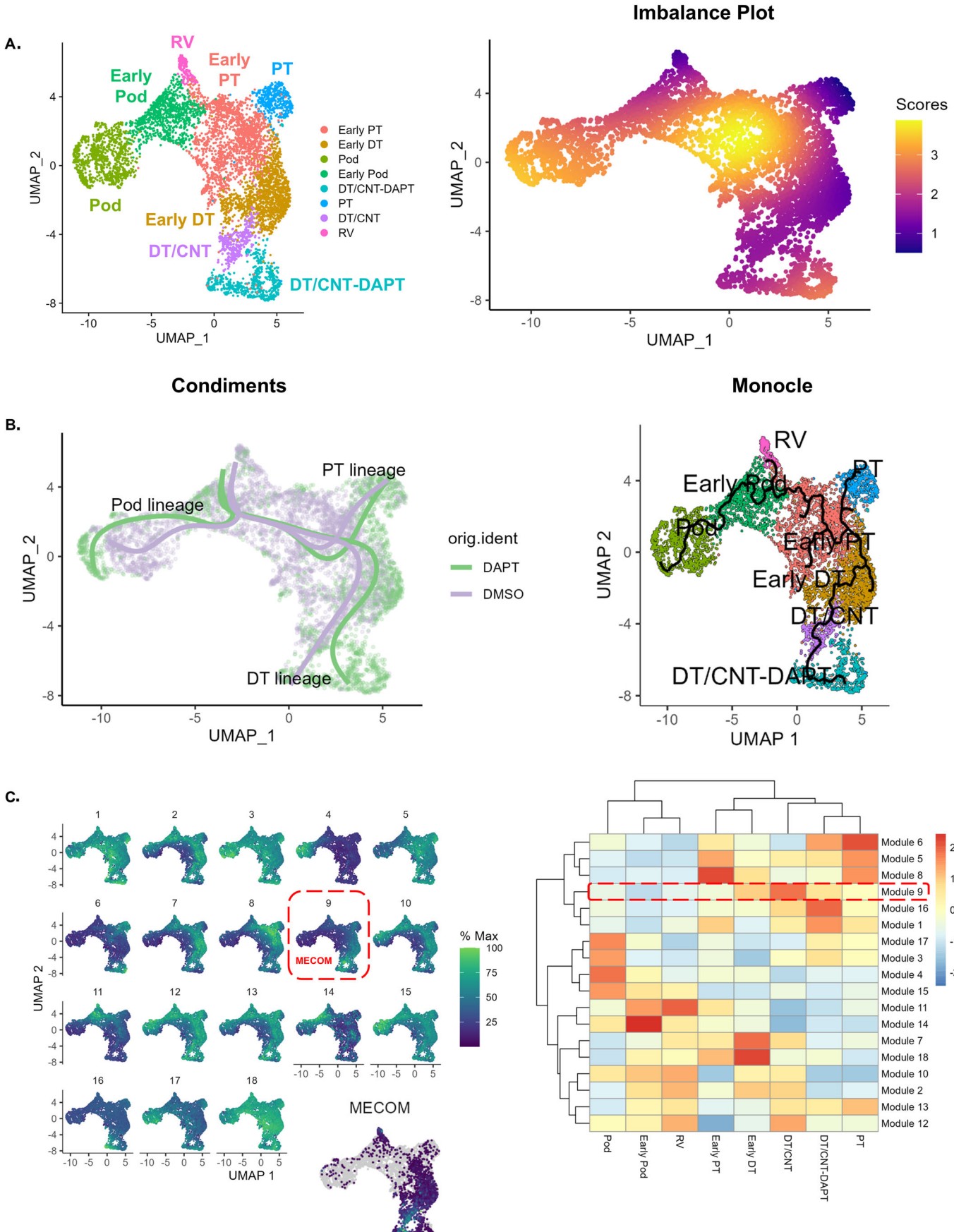

**Figure EV8. Proximal tubule lineage is hampered under Notch inhibition in hFKOs.**

(A) scRNA-seq of hFKOs under Notch inhibition (DAPT) and control (DMSO), and UMAP of the nephrogenic compartment. RV, renal vesicle; PT, proximal tubule; Pod, podocyte; DT, distal tubule; DT/CNT, distal tubule/connecting tubule; DT/CNT-DAPT, distal tubule/connecting tubule enriched under Notch inhibition. Imbalance plot depicting the area in the UMAP with the highest amount of mismatch between control and DAPT, area containing mostly the early PT cluster. (B) Comparison between pseudotime plot from the condiments package and monocle, leading to similar patterns of trajectory inference. (C) Breakdown of trajectory into modules, making it possible to scan for DE genes in certain patterns. Module 9 depicts a pattern of bypass whereby early distal cells circumvent Notch inhibition, putatively through the influence of MECOM.

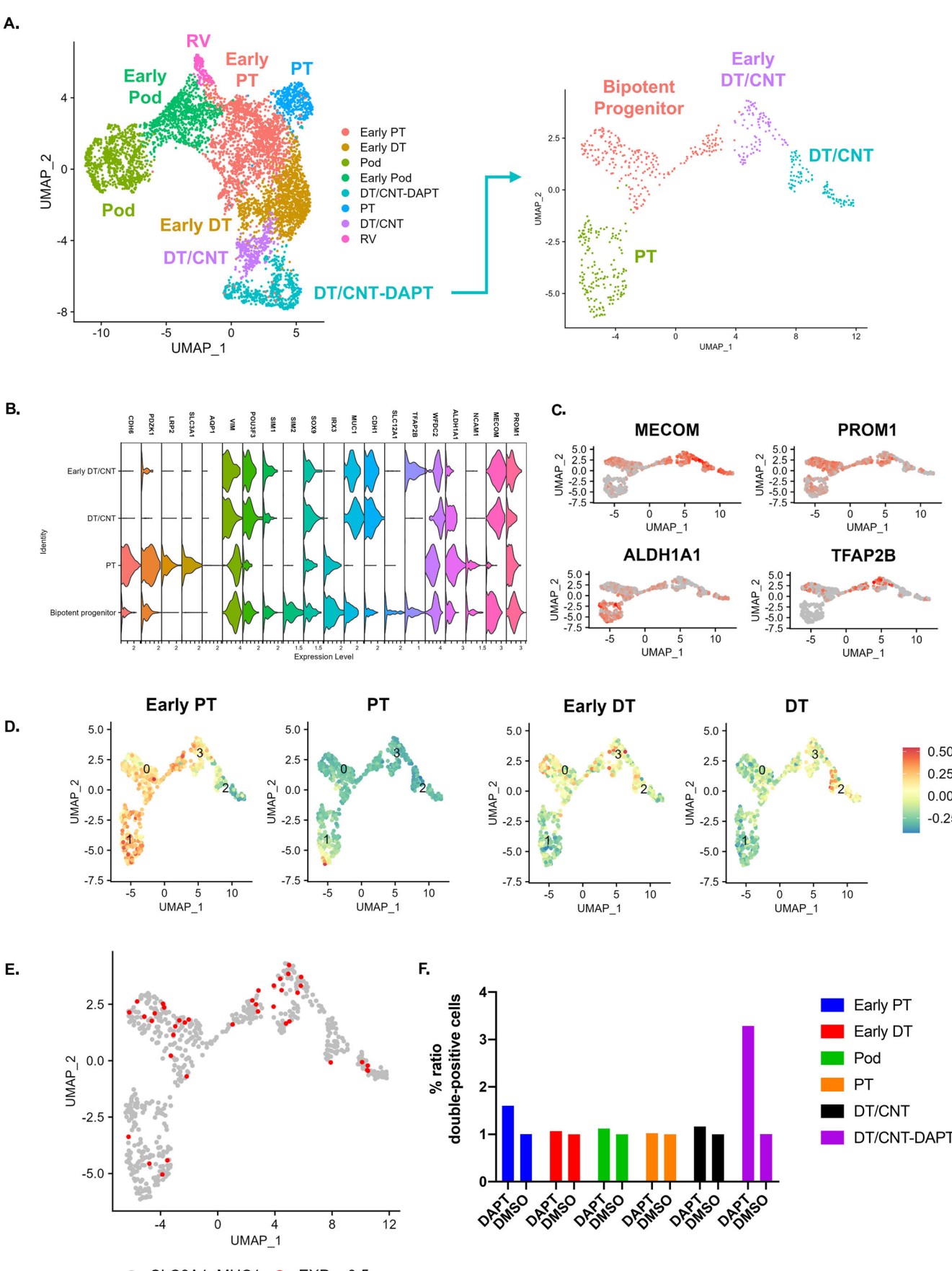

◄ **Figure EV9.   Proximal tubule lineage is hampered under Notch inhibition in hFKOs.**

(A) Sub-setting and re-clustering of the distal tubule/connecting tubule enriched under Notch inhibition (DT/CNT-DAPT) reveals four clusters: bipotent progenitors, early DT/CNT, DT/CNT, and PT. (B) Violin plots of the expression of key markers used to characterize the four sub-clusters in the DT/CNT-DAPT cluster. (C) Feature plot depicting expression patterns of the distal markers *MECOM* and *TFAP2B*, localizing in the distal clusters; *ALDH1A1*, localizing in the PT cluster; and *PROM1* (*CD133*), expressed in all clusters. (D) Feature plot expression heatmap of genes characterizing each lineage, Early PT (*CDH6, HNF1A, IGFBP7*, etc.), PT (*SLC3A1, SLC4A4, SLC3A1*, etc.), early DT (*POU3F3, SOX9, IRX2*, etc.), and DT (*SLC12A3, MUC1, CALB1*, etc.) (E) Double-positive cells detected in the DT/CNT-DAPT cluster, expressing proximal marker *SLC3A1* and distal marker *MUC1*; threshold is set to above 0.5. (F) Percent ratio bar plot of double-positive cells expressing *SLC3A1* and *MUC1* in the various clusters.

