## [Peer Review File · The EMBO Journal]

Human fetal kidney organoids model early human nephrogenesis and Notch-driven cell fate

Michael Namestnikov, Osnat Cohen-Zontag, Dorit Omer, Yehudit Gnatek, Sanja Goldberg, Thomas Vincent, Swati Singh, Yair Shiber, Tal Rafaeli Yehudai, Hadas Volkov, Dani Folkman Genet, Achia Urbach, Sylvie Polak-Charcon, Igor Grinberg, Naomi Pode Shakked, Boaz Weisz, Zvi Vaknin, Benjamin Freedman, and Benjamin Dekel

Corresponding author: Benjamin Dekel (Binyamin.Dekel@sheba.health.gov.il)

Review Timeline:

Submission Date:	6th May 25
Editorial Decision:	28th May 25
Additional Correspondence:	8th Jun 25
Revision Received:	16th Jun 25
Accepted:	1st Jul 25

Editor: Daniel Klimmeck

Transaction Report:

Please note that the manuscript was previously reviewed at another journal and the reports were taken into account in the decision making process at The EMBO Journal. Since the original reviews are not subject to EMBO Press' transparent review process policy, the reports and author response cannot be published.

Dear Dr Dekel,

Thank you again for the submission of your amended manuscript (EMBOJ-2025-121273) to The EMBO Journal. Please accept again my apologies for the unusual protraction due to delayed expert input. We have carefully assessed your manuscript and the point-by-point response provided to the referee concerns that were raised during review at a different journal. In addition, and as mentioned before, we decided to involve an arbitrating expert to evaluate the revised version of your work, with respect to technical robustness, conceptual advance and overall suitability of your work for publication in The EMBO Journal.

As you will see from the arbitrating comment enclosed below, the advisors are favour of the work stating the interest and value of your model and results and therefore supportive of publication at The EMBO Journal. They also offer constructive input on how to improve the manuscript for a final revision.

We are thus pleased to inform you that we can offer to swiftly move forward towards acceptance of this work at The EMBO Journal.

Please consider below inout carefully and revise your manuscript accordingly by additional data or textual adjustments and introducing caveats where appropriate.

Also, we now need you to take care of a number of minor issues related to formatting and data annotation, which I will share shortly in a separate message, together with additional changes and requests by our production team and information regarding Source Data provision.

Please submit a revised version of the manuscript using the link enclosed below, addressing the advisor's comments.

As you might have seen on our web page, every paper at the EMBO Journal now includes a 'Synopsis', displayed on the html and freely accessible to all readers. The synopsis includes a 'model' figure as well as 2-5 one-short-sentence bullet points that summarize the article. I would appreciate if you could provide this figure and the bullet points.

Thank you again for giving us the chance to consider your manuscript for The EMBO Journal, I look forward to hearing from you and receiving your final revised version of the manuscript.

Kind regards,

Daniel Klimmeck

EMBOJ-2025-121273

Arbitrating advisors' comment:

'We have reviewed the manuscript, the reviewers' comments, and the authors' response. We agree with Reviewer 3 that some of the claims are too strongly worded and that certain conclusions may be overinterpreted. We also understand your dilemma regarding the suitability of this study for EMBO.

Here is our perspective:

The authors claim to have established a fetal kidney organoid protocol that enables the study of kidney development in vitro. The use of fetal tissue to generate organoids or organotypic cultures is, in itself, a valuable approach to model early development. Examples such as explant cultures on transwells are widely used, although their major limitation is rapid tissue degradation. In this light, a robust fetal organoid protocol would indeed be a significant advancement.

The authors dissociate fetal kidney tissue and allow it to re-aggregate in Matrigel. Most of their analyses are performed on passages 0, 1, and 2. Later passages are assessed using histological staining and bulk RNA-sequencing. The early-passage organoids recapitulate key aspects of fetal kidney development. However, in later passages, the cultures are dominated by mature epithelial cells, which the authors interpret as evidence of differentiation. In our view, the data suggest instead that the culture conditions selectively support the outgrowth of epithelial populations—a phenomenon we have also observed in our hands. Their early-passage cultures are reminiscent of the MicroOrganoSpheres developed by Xilis, in which dissociated tissues are encapsulated in Matrigel microdroplets.

In its current form, we believe the manuscript overstates its conclusions, particularly regarding the claim of establishing a fetal kidney organoid model. That said, the system presented is still a valuable tool for studying early human kidney development. If the authors temper their claims accordingly, we believe the manuscript could be suitable for publication in EMBO.

Fundamentally, this work describes a short-term culture system derived from fetal kidney tissue that can be used to explore early developmental processes in more detail. The observation that dissociated fetal kidney tissue can be viably frozen and later used to generate organotypic cultures is a noteworthy and useful finding.

We would suggest that the authors refer to these cultures not as "organoids" but rather as "organotypic cultures," which more accurately reflects the nature and duration of the system they describe.'

Dear Dr Dekel

Further to below, I am sharing enclosed the mentioned additional formatting requirements for your article.

Please let us know any time should you have questions related.

We look forward to your resubmission.

Best regards,

Daniel Klimmeck

>> Please add up to five keywords to your study.

>> Provide the main manuscript text as .docx file.

>> Limit the abstract to maximally 175 words.

>> Limit the title to maximally 100 characters (incl spaces).

>> Author Contributions: Remove the author contributions information from the manuscript text. Note that CRediT has replaced the traditional author contributions section as of now because it offers a systematic machine-readable author contributions format that allows for more effective research assessment. and use the free text boxes beneath each contributing author's name to add specific details on the author's contribution.

More information is available in our guide to authors.

>> Adjust the title of the 'Declaration of Interests' section to 'Disclosure and Competing Interests Statement'.

>> Provide a completed Author Checklist.

>> Section order should be corrected as follows: title page with complete author information, abstract, keywords, introduction, results, discussion, methods, data availability section, acknowledgements, disclosure and competing interests statement, references, main figure legends, tables, expanded figure legends.

>> "Summary" should be renamed "Abstract"

- > "Materials and Methods" should be renamed "Methods"
- > Figures in separate files: main figures should be removed from the manuscript and uploaded as individual, high resolution figure files. The supplementary figures should be renamed "Figure EV1" - EV9, their legends should be removed from the figures and added to the manuscript text, after the main figure legends and under the heading "Expanded View Figure Legends". The suppl. figures should also be uploaded as high-resolution figure files.
- > References: please adjust reference format to EMBO Journal format, 10 authors et al. .
- > Figure callouts: please recheck callouts for Fig 3A in the main text.
- > Please add a Reagents and Tools table to the Methods section, as a separate file using the existing template in the Guide For Authors, listing key reagents, experimental models, software and relevant equipment.

Overall, we suggest classifying your study as a 'Methods' article. Accordingly, Please describe the new original methods developed using a step-by-step protocol format with bullet points and notes on "tricky steps" in the procedure. We strongly recommend this 'protocol' format for novel methodologies and procedures that will be relevant for readers to use in their own studies. Using the protocol format is optional unless otherwise specified by the editor. The protocol format does not need to be applied throughout the Methods section.

Please compare also our Guide-to-Authors instructions:

<https://www.embopress.org/page/journal/14602075/authorguide#structuredmethods>

- > Add a separate 'Statistical Analysis' section to the Methods part, detailing the algorithms and statistical tests applied.
- > Dataset EV legends: the movie needs a legend added in a ZIP file; the nomenclature should be corrected to "Movie EV1" in the file and in the manuscript text.
- > Please provide source data for the study as to the separate request e-mail. Source data should be uploaded as one (zipped) file per figure.
- > Funding: ensure that the complete list of funders is entered in both the Acknowledgements and in our system. A number of funders is currently missing in our system. Please add project numbers where available.
- > Data availability section: please enter a Data availability section into the manuscript, detailing the deposition of the single-cell RNAseq data. Please provide a URL for the dataset and ensure privacy is released and the data is public.
- > Biorender: Please remove the sentence from the Acknowledgments and add it to the Methods section instead, following this format:
Graphics:
(some of the... OR Figure #... OR synopsis) Graphics were created with BioRender.com.
- > Please recheck references for the bioRxiv entry Bezieux et al. (2021) and update the citation if in the meantime published as regular article.
- > Consider additional changes and comments from our production team as indicated below:

-Figure legends:

1. Please note that the exact p values are not provided in the legend of figure 5B
2. Please indicate the statistical test used for data analysis in the legends of figure 5B
3. Please note that scale definition is missing for figures 5C-F

Response to EMBO's arbitrating advisor comments:

We would like to thank the arbitrating advisors for their thoughtful and constructive comments. We appreciate the time and care taken to evaluate our manuscript, as well as the balanced perspective provided on the strengths and limitations of our study. We would be glad to revise the manuscript accordingly and believe that the suggested changes will help improve the clarity and impact of our study.

We acknowledge the limitation of our culture system in terms of the difficulty of performing lineage tracing in primary cultures, thus we cannot confirm with a certainty that the kidney progenitors which are abundantly present in early passages are differentiating into mature kidney epithelium. Importantly, there is evidence in the performed bulk-RNA comparative analysis between P0-P2-P6 that there's a shift in expression of early developmental genes towards more mature expression profiles. Moreover, as the advisor noted, the appearance of more mature elements over time may represent culture preferences selecting for mature cell types that were there in the beginning rather than maturation events. We acknowledge this point and will incorporate it into the discussion (**line 552**). Notably, prospective isolation of NCAM1+ epithelial progenitors from P2 human fetal kidney organoids (hFKOs), and re-culturing gave rise to early NCAM1+PAX2+ and NCAM1+EpCAM1+ more mature epithelial precursors (indicative of renewal of the NCAM1+ population) as well into differentiated population devoid of NCAM1, e.g. LTL+EMA+HNF1B+ proximal and distal nephron epithelium. This indicates processes which cannot be attributed solely to selection by culture conditions. Thus, even though we believe that our culture system allows differentiation and maturation we agree with the advisors that our claims could be "toned down" by further explaining culture selection bias in the discussion section (**line 552**) and the lack of lineage tracing methods to fully comprehend the maturation trajectories in a complex and heterogenous culture system such as the hFKOs.

The long-term culture element of the hFKOs is relative to the current prominent PSC-derived kidney organoids (PSC-KOs) which are used in the field. Today, these cultures support the maturation and growth of 1st trimester embryonic kidney-like structures for a maximum of 30 days. Towards the 30-day mark of the protocols, there's usually a shift towards dedifferentiation causing of decline in expression of kidney markers as well as viability. In contrast, our hFKOs continue to express early developmental markers even in P2, ~3 months after the start of the culture; importantly, both genomic data and immunostaining unequivocally show the persistence of epithelial progenitors for 3 months in culture. We think that this is a substantial improvement and a key feature in the hFKO culture system. If "long-term" is more suited to the timeframes offered by adult kidney organoids, also known as "tubulods", which can be potentially cultured indefinitely, we suggest using "prolonged" throughout the manuscript to clarify that hFKOs can be cultured for longer than PSC-KOs.

We recognize that in recent years, the term "organoids" has been used very loosely, often employed in contexts that do not fully meet the original definition. Our colleagues and us believe that organoids should originate from PSCs (iPSCs or ESCs) or adult stem cells/progenitors, self-organize into complex structures resembling the tissue of origin, express tissue specific markers and function and be able to be expanded and passaged over time. We believe that the human fetal kidney 3D biological entities which we are growing in vitro, align with the requirements of being called organoids. hFKOs are derived from dissociated fetal kidney tissue. The tissue is fully dissociated into single cells rather than small tissue fragments or slices and in that regard has no features of an organotypic culture. It is then cultured in ECM where single cells self-organize into 3D, polarized epithelial structures which react to Forskolin stimulation. The cultures recapitulate nephron

segment identities, including nephron progenitor cells (NPCs), renal vesicles (RVs), proximal/distal tubules, and podocytes, and even contain stromal and endothelial populations. Bulk and single-cell transcriptomics confirm that hFKOs undergo a developmental trajectory, matching fetal kidney development in vivo. hFKOs can be propagated for months (up to P6), maintaining expression of early developmental genes in early passages (P2), confirmed by functional isolation of NCAM1+ epithelial progenitors and their subsequent growth into de-novo organoids.

The Clevers group has referred to tubuloids as adult kidney organoids (AKOs), even though they often exhibit a cystic phenotype (which is pathological in healthy kidneys) and downregulate key nephron segment markers. A recent paper utilized organotypic culture of large pieces of brain tissue and referred to the explant structures as “Human fetal brain organoids” <https://doi.org/10.1016/j.cell.2023.12.012>.

To conclude, we believe that hFKOs meet the structural, cellular, and functional criteria to be termed “organoids”, particularly in early passages where they recapitulate fetal nephron development, self-organize in 3D, and retain multipotent progenitors. At the same time, their derivation from primary tissue, even though they are dissociated to single cells, could also support their classification as “organotypic cultures.” The dual terminology reflects both the biological complexity and the practical utility of the model. Hence, we suggest using both terms for easy readability and understanding by the readers, which will associate organoids as 3D structures resembling the tissue of origin in-vitro while explaining the organotypic culture in the introduction section of the manuscript (**line 98**).

We have carefully addressed all editorial and formatting requirements listed in the revision instructions. This includes limiting the abstract and title length, reorganizing manuscript sections, updating figure files and legends and adjusting reference formatting. We have also incorporated a Data Availability section, added missing funding details, updated author contributions using the CRediT taxonomy, and ensured compliance with all technical specifications, including the Reagents and Tools table and statistical analysis descriptions. We are currently experiencing technical difficulties uploading our datasets to the GEO database, likely due to a communication issue with their servers. We have attempted the upload from multiple computers and institutional networks without success. The Data Availability section has been included in the manuscript, and we are actively working to resolve the issue to complete the deposition.

Dear Dr Dekel,

Thank you for submitting the revised version of your manuscript. I have now evaluated your amended manuscript and concluded that the remaining minor concerns have been sufficiently addressed.

I am thus pleased to inform you that your manuscript has been accepted for publication in the EMBO Journal.

On a different note, I would like to alert you that EMBO Press offers a format for a video-synopsis of work published with us, which essentially is a short, author-generated film explaining the core findings in hand drawings, and, as we believe, can be very useful to increase visibility of the work. Please see the following link for representative examples and their integration into the article web page:

<https://www.embopress.org/doi/full/10.15252/emj.2019103932>

Best regards,

Daniel Klimmeck

Daniel Klimmeck, PhD
Senior Editor
The EMBO Journal
EMBO
Postfach 1022-40
Meyerhofstrasse 1
D-69117 Heidelberg
contact@embojournal.org